# Unveiling the domain-specific and RAS isoform-specific details of BRAF kinase regulation

Tarah Elizabeth Trebino[1], Borna Markusic[1,2], Haihan Nan[1,3], Shrhea Banerjee[1], Zhihong Wang[1]*

[1]Rowan University, Glassboro, United States; [2]Max Planck Institute of Biophysics, Frankfurt am Main, Germany; [3]School of Laboratory Medicine and Life Science, Wenzhou Medical University, Wenzhou, China

*For correspondence:
wangz@rowan.edu

Competing interest: The authors declare that no competing interests exist.

**Abstract** BRAF is a key member in the MAPK signaling pathway essential for cell growth, proliferation, and differentiation. Mutant BRAF is often the underlying cause of various types of cancer and mutant RAS, the upstream regulator of BRAF, is a driver of up to one-third of all cancers. BRAF interacts with RAS and undergoes a conformational change from an inactive, autoinhibited monomer to an active dimer, which propagates downstream signaling. Because of BRAF's complex regulation mechanism, the exact order and magnitude of its activation steps have yet to be confirmed experimentally. By studying the inter- and intramolecular interactions of BRAF, we unveil the domain-specific and isoform-specific details of BRAF regulation through pulldown assays, open surface plasmon resonance (OpenSPR), and hydrogen-deuterium exchange mass spectrometry (HDX-MS). We demonstrate that the BRAF specific region (BSR) and cysteine rich domain (CRD) play a crucial role in regulating the activation of BRAF in a RAS isoform-specific manner. Moreover, we quantified the binding affinities between BRAF N-terminal and kinase domains (KD) to reveal their individual roles in autoinhibition. Our findings also indicate that oncogenic BRAF-KD$^{D594G}$ mutant has a lower affinity for the N-terminal domains, implicating that pathogenic BRAF acts through decreased propensity for autoinhibition. Collectively, our study provides valuable insight into the activation mechanism of BRAF kinase to guide the development of new therapeutic strategies for cancer treatment.

## eLife assessment

This manuscript describes **useful** information on the interactions of the BRAF N-terminal regulatory regions (CRD, RBD and BSR) with the C-terminal kinase domain and with the upstream regulators HRAS and KRAS. The authors provide **solid** evidence that the BRAF BSR domain may negatively regulate RAS binding and propose that the presence of the BSR domain in BRAF provides an additional layer of autoinhibitory constraints. The data will be of interest for researchers in the RAS/RAF and general kinase regulation fields.

## Introduction

The RAF family, composed of A-, B-, and CRAF (Raf1) in mammalian cells, are serine/threonine kinases that function to modulate cell growth and differentiation (*Lavoie and Therrien, 2015*). The RAF family is a key component in the RAS-RAF-MEK-ERK (MAPK) signaling cascade. Upon extracellular stimulation, the GTPase protein, RAS, becomes activated with the aid of GEFs to adopt the GTP-bound active form (*Malumbres and Barbacid, 2003*). Subsequently, RAF is activated by a number of events

such as interacting with active RAS (*Zhang et al., 1993*; *Vojtek et al., 1993*), relieving autoinhibition (*Cutler et al., 1998*; *Tran and Frost, 2003*), translocating to the membrane, and forming dimers (*Rajakulendran et al., 2009*). Active RAF then phosphorylates and activates MEK, which in turn phosphorylates and activates ERK (*Crews and Erikson, 1992*). Finally, activated ERK translocates to the nucleus, where it regulates various cell processes (*Yoon and Seger, 2006*). A significant number of cancers are linked to mutations of MAPK components, with RAS being mutated in 10–30% of all human cancers (*Prior et al., 2012*). BRAF mutations are the cause of roughly 8% of cancers (*Davies et al., 2002*). In addition, germline mutations in RAS and RAF lead to RASopathies—a variety of genetic diseases that cause developmental disorders such as facial deformation and cardiovascular deficiencies (*Tajan et al., 2018*).

Targeting RAS in cancer treatments is challenging because of its compact shape, shallow cavities on its smooth surface, and extremely high binding affinity for GTP. Currently, despite decades of research, RAS only has two FDA-approved inhibitors that work by covalently attaching to the oncogenic G12C mutation of KRAS (*Canon et al., 2019*; *Fell et al., 2020*). RAS has four isoforms (NRAS, HRAS, KRAS4A, and KRAS4B) and numerous oncogenic mutations other than the G12C mutation. Furthermore, BRAF mutations are categorized into three classes based on RAS and dimer dependency (*Yao et al., 2015*). The most common BRAF mutation is the V600E substitution, a class 1 RAS- and dimer-independent mutation (*Davies et al., 2002*). Class 2 mutants are RAS-independent but dimer-dependent and class 3 mutants are both RAS- and dimer-dependent (*Yao et al., 2015*). However, FDA-approved inhibitors, vemurafenib, dabrafenib, and encorafenib, are limited to class 1 mutations that typically signal as a monomer. Since class 2 and 3 mutants and wild-type BRAF signal as dimers, the current FDA-approved inhibitors promote the paradoxical activation phenomenon, binding to only one protomer and allosterically activating the other, in cells containing these mutant or wild-type proteins (*Yao et al., 2015*; *Hatzivassiliou et al., 2010*; *Poulikakos et al., 2010*). These limitations arise from knowledge gaps in RAF regulation and shortcomings in current drug treatments. Therefore, elucidating the details of the RAS-RAF interaction and the regulatory events surrounding it are essential for advancing the field and designing new therapies for the biological system that is prevalent in human cancers.

The RAF kinase family is comprised of three main conserved regions (CR1, CR2, and CR3), each with specific, non-overlapping functions in RAF regulation (*Figure 1*). The CR1 contains the RAS binding domain (RBD) and the cysteine rich domain (CRD). The CR2 is a flexible linker region that harbors a RAF phosphorylation site and binding site for 14-3-3, which helps maintain RAF in its autoinhibited state (*Park et al., 2019*). The CR3 contains the catalytic or kinase domain (KD), which is important for RAF dimerization and phosphorylation of MEK substrates (*Rajakulendran et al., 2009*; *Kyriakis et al., 1992*). The N-terminal region, also known as the regulatory region, is an essential feature of RAF architecture which regulates RAF activity through the concerted actions of the domains in this region: the RBD, CRD, and BRAF specific region (BSR) in BRAF.

The CRD has multifaceted roles in membrane recruitment, RAS interaction, as well as RAF autoinhibition. RBD-CRD interactions with RAS and direct interaction with anionic phospholipids anchor RAF to the membrane for RAF activation (*Roy et al., 1997*; *Ghosh et al., 1994*; *Fischer et al., 2007*; *Li et al., 2018*). While the RBD is the primary domain involved in the strong nanomolar affinity interaction with RAS, a number of studies have also shown that the CRD increases the affinity of CRAF for HRAS, even though the CRD has a weaker micromolar affinity on its own (*Brtva et al., 1995*; *Hu et al., 1995*; *Williams et al., 2000*). Early research showed that CRD interaction with RAS is required for RAF activation (*Roy et al., 1997*; *Hu et al., 1995*). Recently, the crystal structure of KRAS in complex with CRAF-RBD-CRD revealed the previously unknown CRD binding interface. Interactions at the interswitch region and C-terminal helix α5 of KRAS, along with mutagenesis experiments, further solidified the CRD-RAS interaction as necessary for RAF activation (*Tran et al., 2021*). Another structure of the CRAF-RBD-CRD in complex with HRAS, resolved nearly concurrently, also supported the central role of the CRD, in which it is poised to modulate RAS and RAF functionalities because of its location at the base of two RAS protomers (*Cookis and Mattos, 2021*). The cryo-EM structures of autoinhibited BRAF in complex with the regulatory protein 14-3-3 and MEK confirmed the importance of the CRD in negatively regulating catalytic activity through interactions with the BRAF C-terminal KD and 14-3-3 (*Park et al., 2019*; *Martinez Fiesco et al., 2022*). CRD-KD interactions stabilize the inactive monomeric complex, while 14-3-3 blocks the BRAF dimer interface, thereby preventing the activation of KD

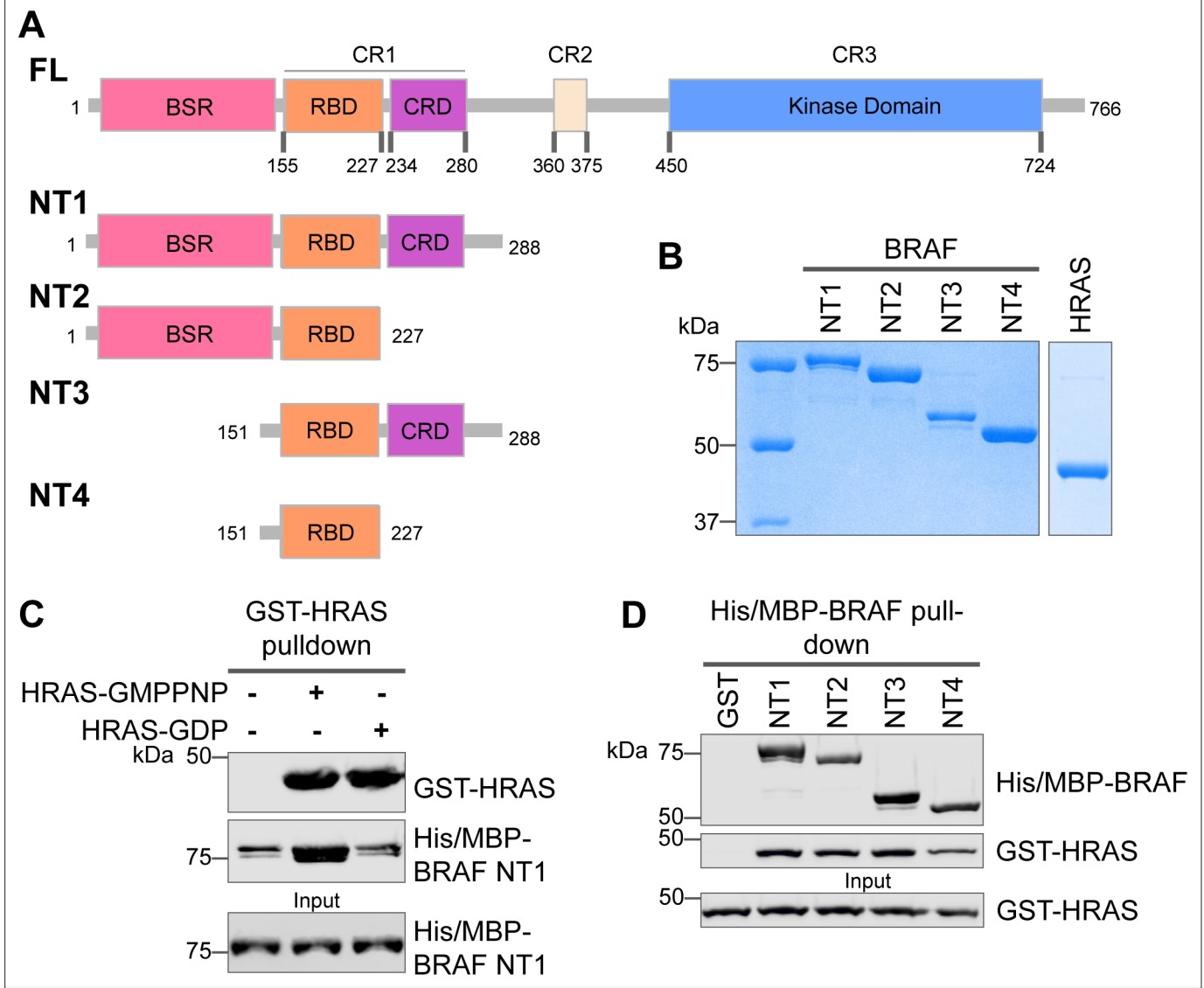

**Figure 1.** Specific purified N-terminal domains are involved in HRAS interactions. (**A**) Diagram of BRAF NT constructs. Top panel is full-length BRAF, followed by proteins NT1–4 expressed in *Escherichia coli* and purified. Not shown: 6xHis/MBP tag on the N-terminal of NT proteins. BSR: BRAF specific region; RBD: RAS binding domain; CRD: cysteine rich domain. (**B**) Coomassie stained gels of purified NT1–4 and GST-HRAS. (**C**) Western blot of HRAS-GMPPNP or HRAS-GDP pulled down on glutathione resin to probe for NT1 binding. (**D**) Western blot of BRAF NT1–4 pulled down on amylose resin to probe for HRAS-GMPPNP binding.

The online version of this article includes the following source data and figure supplement(s) for figure 1:

**Source data 1.** (Panel b) Coomassie stained gels of purified NT1–4 and GST-HRAS.

**Source data 2.** (Panel c) Western blot of HRAS-GMPPNP or HRAS-GDP pulled down on glutathione resin to probe for NT1 binding.

**Source data 3.** (Panel d) Western blot of BRAF NT1–4 pulled down on amylose resin to probe for HRAS-GMPPNP binding.

**Figure supplement 1.** SEC of active GST-HRAS.

(*Park et al., 2019*). By combining these structure results with simulations, a mechanism was proposed: the extraction of the CRD through RAS interaction leads to the activation of RAF (*Tran et al., 2021*; *Martinez Fiesco et al., 2022*). Additionally, the RBD is suggested to be a critical modulator of the transition from monomeric to dimeric RAF complexes due to the steric clash that would occur upon RAS binding (*Martinez Fiesco et al., 2022*). However, none of the structural studies was able to capture the BSR, likely due to its high degree of flexibility.

Differences among RAF isoforms are an important and not yet fully understood distinction in MAPK activation. While many studies have examined the structure and regulation of CRAF, these findings may not translate to all RAF isoforms. BRAF has the highest basal activity compared to ARAF and CRAF and is thus mutated most frequently in cancer (*Marais et al., 1997*; *Emuss et al., 2005*). Furthermore, compared to CRAF-CRD, the BRAF-CRD exhibits increased autoinhibitory activity and membrane binding (*Spencer-Smith et al., 2022*). BRAF and CRAF are suggested to have different preferences for H/K/NRAS, although discrepancy exists among these studies (*Fischer et al., 2007*; *Terrell et al., 2019*). Compared to CRAF, BRAF associates with unmodified HRAS with much higher affinity (*Fischer et al., 2007*). However, in recent BRET studies, BRAF was shown to prefer KRAS over HRAS, whereas CRAF did not differentiate between RAS isoforms (*Terrell et al., 2019*). The BSR was implicated in regulating these distinct binding preferences with RAS isoforms (*Terrell et al., 2019*). Other than this recent study, relatively little research has examined the role of the BSR despite being one of the most noticeable isoform differences, making it an intriguing feature of regulation to study.

The Raf activation process is dynamic and complex, and despite years of research many details remain unclear. While many of BRAF's activation steps are built upon static structures and cell-based site-directed mutagenesis, the order and magnitude of these events has yet to be experimentally validated in vitro. The precise mechanism of how autoinhibition is released upon RAS binding is unknown, as well as the communication between the regulatory domains and the KD. Our current knowledge of the RAS-RAF interaction is derived mainly through characterization of CRAF, and a comprehensive analysis of BRAF-RAS interaction is still missing. BRAF has long been believed to have a distinct regulation mechanism, however, it remains elusive how BRAF differentiates itself from other RAF family members.

Here, we investigate the interactions of BRAF regulatory regions with the C-terminal KD and with upstream regulators, HRAS and KRAS. To our knowledge, we present the first reported $K_D$ values for the N- and C-terminal interactions of BRAF. Our results demonstrate that the CRD plays a primary role in autoinhibitory interactions, while the presence of the BSR increases the affinity for the KD. The RBD is the primary driver of RAS-RAF binding, however the BSR and CRD have allosteric effects that slow the association with HRAS, thus providing isoform specificity toward KRAS. We also show that HRAS binding to BRAF disrupts the N- to C-terminal autoinhibitory interactions and that the oncogenic $BRAF^{D594G}$ can relieve autoinhibition to promote activity. Overall, this comprehensive in vitro study of the BRAF N-terminal region provides new insights into BRAF regulation.

## Results

### Specific purified BRAF N-terminal domains are involved in HRAS interactions

The interactions between RAS and RAF have been well established, occurring primarily between the RAF-RBD region and secondarily between the RAF-CRD region to enhance the affinity for RAS (*Simanshu and Morrison, 2022*). However, the contribution of each regulatory domain to the BRAF activation mechanism is still not completely understood. We hypothesized that studying the differences in HRAS binding to various BRAF N-terminal constructs (NTs) would highlight the role of each domain and their cooperation in fine-tuning BRAF. We purified four different N-terminal BRAF constructs comprising an N-terminal MBP tag and various domains: NT1 (aa 1–288), NT2 (aa 1–227), NT3 (aa 151–288), and NT4 (aa 151–227; *Figure 1A and B*). We also purified GST-tagged full-length HRAS in both active GMPPNP-loaded and inactive GDP-loaded forms. As verified by size exclusion chromatography (*Figure 1—figure supplement 1A*), the GST-tag dimerizes and thus forces HRAS into close proximity to recapitulate physiological conditions (*Simanshu et al., 2023*). After incubating BRAF NT1 with either active or inactive HRAS for 1 hr, we conducted pulldown assays, in which NT1 is captured by amylose beads and probed for HRAS. Our results demonstrate that active HRAS binds to BRAF NT1 with stronger affinity than inactive HRAS, confirming that the purified HRAS protein in both the active form and inactive form, behaves as expected (*Figure 1B and C*). We also found that all NT constructs bind to active HRAS in pulldown assays, suggesting that purified BRAF fragments in vitro recapitulate the physiological protein-protein interactions that occur in cells (*Figure 1D*).

# HDX-MS reveals conformational changes of BRAF N-terminal domains in response to HRAS binding

To further evaluate the specific regions of interaction and conformational changes in the BRAF regulatory domain upon HRAS binding, we performed hydrogen-deuterium exchange mass spectrometry (HDX-MS) experiments with two constructs, NT2 (includes BSR and RBD) and NT3 (includes RBD and CRD). Both constructs were incubated with and without active HRAS in $D_2O$ buffer for set labeling reaction times (NT2: 20 s, 30 s, 60 s, 5 min, 10 min, 90 min, 4.5 hr, 15 hr, and 24 hr at RT; NT3: 2 s, 6 s, 20 s, 30 s, 60 s, 5 min, 10 min, 30 min, 90 min, 4.5 hr, 15 hr, 45 hr, and 24 hr at RT), injected through a pepsin column for digest, and analyzed for deuterium uptake through mass spectrometry. Of the two approaches that exist within the field, we followed the practice of performing exchange reactions across a broad range of labeling time points (much more than four orders of magnitude) to assign data significance rather than multiple replicates of a few time points (*Mayne et al., 2011*; *Kan et al., 2019*; *Ye et al., 2019*; *Ye et al., 2020*). Both constructs have multiple overlapping peptides for almost all residues and good sequence coverage for NT2 and NT3 (*Figure 2—figure supplement 1*). The resulting peptide time plots display rate changes of deuterium exchange across the wide range of labeling time points of peptides from NT2 and NT3 in $D_2O$. A complete set of the time-dependent deuterium uptake plots for NT2 apo- and HRAS-bound and NT3 apo- and HRAS-bound are presented in *Supplementary file 1*. Data are displayed as the uncorrected deuterium uptake (no back exchange corrections) since maximal labeling (100% D uptake) was not demonstrated for the control apo-proteins, in which labeling reactions were performed for 24 hr at RT and quenched at pH 2.4. A trend of four or more overlapping peptides with varying charge states within a sequence range was considered high confidence of whether binding of HRAS has occurred to induce a rate change in deuterium exchange (*Mayne et al., 2011*; *Kan et al., 2019*; *Ye et al., 2019*; *Ye et al., 2020*).

Peptides from HDX-MS experiments with NT2 and HRAS have decreased rate of deuterium exchange encompassing amino acids 174–188 of BRAF, indicating that residues in the RBD interact directly with HRAS (*Figure 2A*). Similarly, peptides from experiments with NT3 and HRAS have decreased rate of deuterium exchange in a broader region of amino acids 158–188, corresponding to a large portion of the RBD (*Figure 2C*). Since the full-length structure of BRAF is still unresolved, we applied the AlphaFold Protein Structure Database for a model of BRAF to display the HDX-MS results on the N-terminal domains (*Jumper et al., 2021*; *Varadi et al., 2022*). The BRAF model (AF-P15056-F1; from UniProt ID: P15056) has high intra-domain confidence based on the pLDDT score, but much weaker inter-domain confidence based on the predicted aligned error. We use the structure confidently as a model to display the H-D exchange differences, however, make no claims to the overall BRAF conformation related to individual domain positioning. Additionally, the RBD features of the BRAF model overlay similarly to resolved structures of the BRAF-RBD region in complex with HRAS (PDB ID: 4G0N; *Fetics et al., 2015*), which adds to our confidence in using the structure as a model. We mapped the change in deuterium exchange on the AlphaFold BRAF structure, with red regions indicating a slower exchange rate (*Figure 2B and D*). These regions of slower deuterium exchange lie within the expected binding interface of BRAF-RBD and include the critical Arg188 residue (R89 in CRAF) for RAS binding (*Tran et al., 2021*; *Fabian et al., 1994*). Our results are consistent with the current model that the RBD is the main region of interaction between BRAF and HRAS. These results further verified that the protein-protein interactions we captured here are physiologically relevant, as our results match with those in the context of mammalian or insect cell expression systems (*Fischer et al., 2007*; *Tran et al., 2021*), and that the proteins we purified from *E. coli* recapitulate the key elements of RAF regulation.

Peptides located outside of the RBD binding region do not display any differential rate decrease when HRAS is present (*Figure 2A and C*). Although a few peptides within the BRAF-CRD region show slower exchange from HDX-MS experiments with NT3 and HRAS, this phenomenon does not appear to occur in the majority of peptide fragments (*Figure 2—figure supplement 2*). Therefore, we cannot conclusively demonstrate interactions with HRAS around the CRD region. However, this may be due to the reported micromolar affinity for the CRD domain's interaction, which is close to the detection limit for HDX-MS studies (*Williams et al., 2000*).

Interestingly, we observed that peptides in the BSR of NT2 exchange deuterium at a faster rate when bound to HRAS, indicating that HRAS binding induces an opening of NT2 and decreased structural rigidity (*Figure 2A*). The peptides with faster exchange correspond to BRAF residues 49–64, and

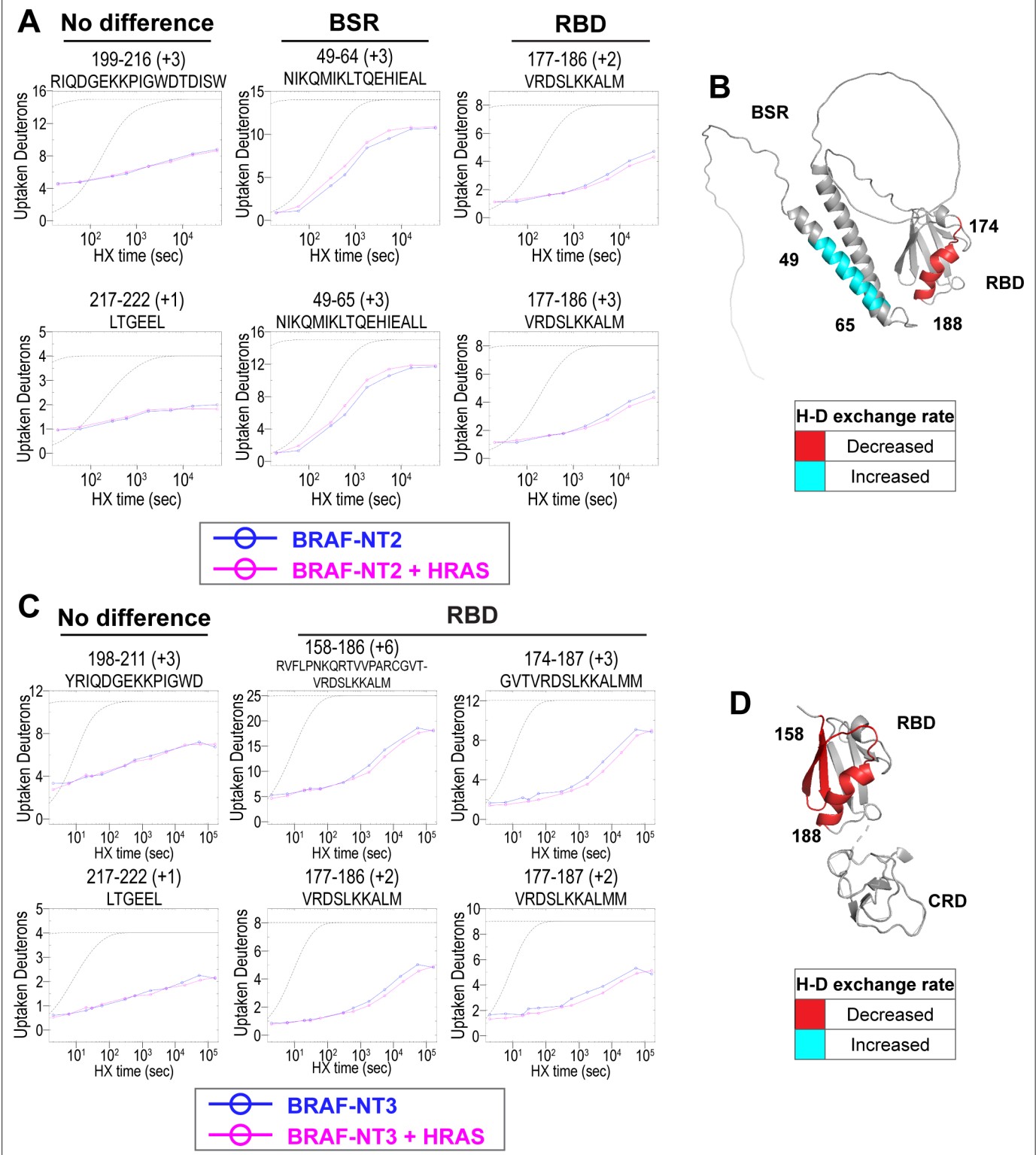

**Figure 2.** Hydrogen-deuterium exchange mass spectrometry (HDX-MS) reveals conformational changes of BRAF N-terminal domains in response to active HRAS binding. (**A, C**) Representative NT2 (**A**) and NT3 (**C**) peptides identified from HDX-MS in the absence (blue) and presence (pink) of HRAS. Peptides with 'no difference' in H-D exchange rate are consistent in both NT2 and NT3. Presented peptide plots displaying differences in H-D exchange rate are representative of a trend of at least four multiple overlapping peptides in the BRAF specific region (BSR) and/or RAS binding domain (RBD). Gray dotted lines represent the theoretical exchange behavior for specified peptide that is fully unstructured (top) or for specified peptide with a

*Figure 2 continued on next page*

*Figure 2 continued*

uniform protection factor (fraction of time the residue is involved in protecting the H-bond) of 100 (lower). (**B, D**) Deuteron uptake differences of NT2 (**B**) and NT3 (**D**) mapped on the predicted BRAF AlphaFold structure, where deuterium exchange is decreased (red) or increased (cyan).

The online version of this article includes the following source data and figure supplement(s) for figure 2:

**Source data 1.** (Panel a) Raw data used to plot curves with EXMS2.

**Source data 2.** (Panel c) Raw data used to plot curves with EXMS2.

**Figure supplement 1.** Stripe plots for peptides identified in BRAF NT2 (**A**) and BRAF NT3 (**B**).

**Figure supplement 2.** Peptides from NT3 in the cysteine rich domain (CRD) region.

**Figure supplement 3.** Peptides from NT2 in the BRAF specific region (BSR) (amino acids 82–99).

while coverage of peptides corresponding to BRAF 82–99 is not as strong, this region also displays some acceleration of exchange (*Figure 2A*; *Figure 2—figure supplement 3*). Due to the shape of the curve, which follows relatively parallel to the theoretical exchange curve, it is reasonable to infer that the BSR and RBD fold to make contacts in NT2 apo. In contrast, HRAS binding at the RBD breaks these contacts and causes a concerted opening as a unit in the BSR. We denote faster deuterium exchange on the AlphaFold BRAF structure with cyan regions (*Figure 2B*). AlphaFold predicts two alpha-helices in the BSR, while the rest of the domain is highly unstructured, which resembles an X-ray structure resolving the BSR (PDB ID: 5VYK) from *Lavoie et al., 2018*. The two alpha-helices of the BSR are the same regions where BRAF NT2 residues display greater structural flexibility when bound to HRAS (*Figure 2B*), which suggests that conformational changes in the N-terminal region occur in the presence of HRAS.

## BSR in conjunction with the CRD reduces binding affinity for HRAS

To further reveal the roles of each domain, we measured the binding affinities of HRAS to each NT construct through OpenSPR experiments. In all experiments, we immobilized His-tagged NTs to a Ni-NTA sensor and flowed over HRAS in the OpenSPR with a flow rate of 30 µL/min. Maltose bind protein (MBP) is immobilized on the OpenSPR reference channel, which accounts for any non-specific binding or for impacts to the native protein-protein interactions that may result from the presence of tags. Kinetic analysis is performed on the corrected binding curves, which subtracts any response in the reference channel. BRAF NT2, NT3, and NT4 bind to HRAS with nanomolar affinity ($K_D$ = 7.5 ± 3.5 nM, 22±11 nM, and 19±11 nM, respectively [mean ± standard deviation]; *Figure 3B–E*). It is noteworthy that the $K_D$ of NT2 from our study ($K_D$ = 7.5 nM) is similar to the previously reported $K_D$ for BRAF residues 1–245 (BSR+RBD) purified from insect cells ($K_D$ = 11 nM), further confirming that post-translational modifications do not affect the binding affinity of this interaction (*Fischer et al., 2007*). Surprisingly, we were unable to observe binding with NT1 even under many varying conditions, changes in experimental design, and new protein preparations (*Figure 3A*). Since the interaction was captured by pulldowns (*Figure 1D*) but not OpenSPR, we investigated whether the interaction time is involved in this discrepancy. We therefore immobilized NT1 and flowed over HRAS at a much slower flow rate (5 µL/min), during which we saw minimal but consistent binding (*Figure 3—figure supplement 1A*). The low response and long time frame of each injection, however, makes the dissociation constant ($K_D$) unmeasurable and incomparable to our other NT-HRAS OpenSPR results. We propose that the conformation of the whole N-terminal region, when BSR, RBD, and CRD are together, prevents rapid association with HRAS. As BRAF truncated to include only the BSR and RBD domains or BRAF truncated to include only the RBD and CRD still exhibits a robust binding affinity, the co-existence of the CRD and the BSR in NT1 introduces distinctions in binding behavior. These results, together with HDX-MS, suggest that the BSR negatively regulates the interaction between HRAS and BRAF, likely in conjunction with the CRD, by blocking the RAS binding surface.

To further validate the slower association rate of BRAF NT1 to HRAS, we performed time-dependent competition pulldowns to compare the association rates of NT1 and NT3 to HRAS (*Figure 3F and G*). NT1 and active HRAS were incubated for 1 hr, followed by the subsequent addition of NT3 for 5, 15, and 30 min. We observed that after 15 min, NT3 started to associate with HRAS, which then proceeded with a time-dependent increase in association (*Figure 3F*). In contrast, when NT3 and HRAS were first incubated for 1 hr, and NT1 was subsequently added for 5, 15, and 30 min, NT1 did not show any association with HRAS (*Figure 3G*). Time-dependent pulldowns of HRAS and NT1 alone

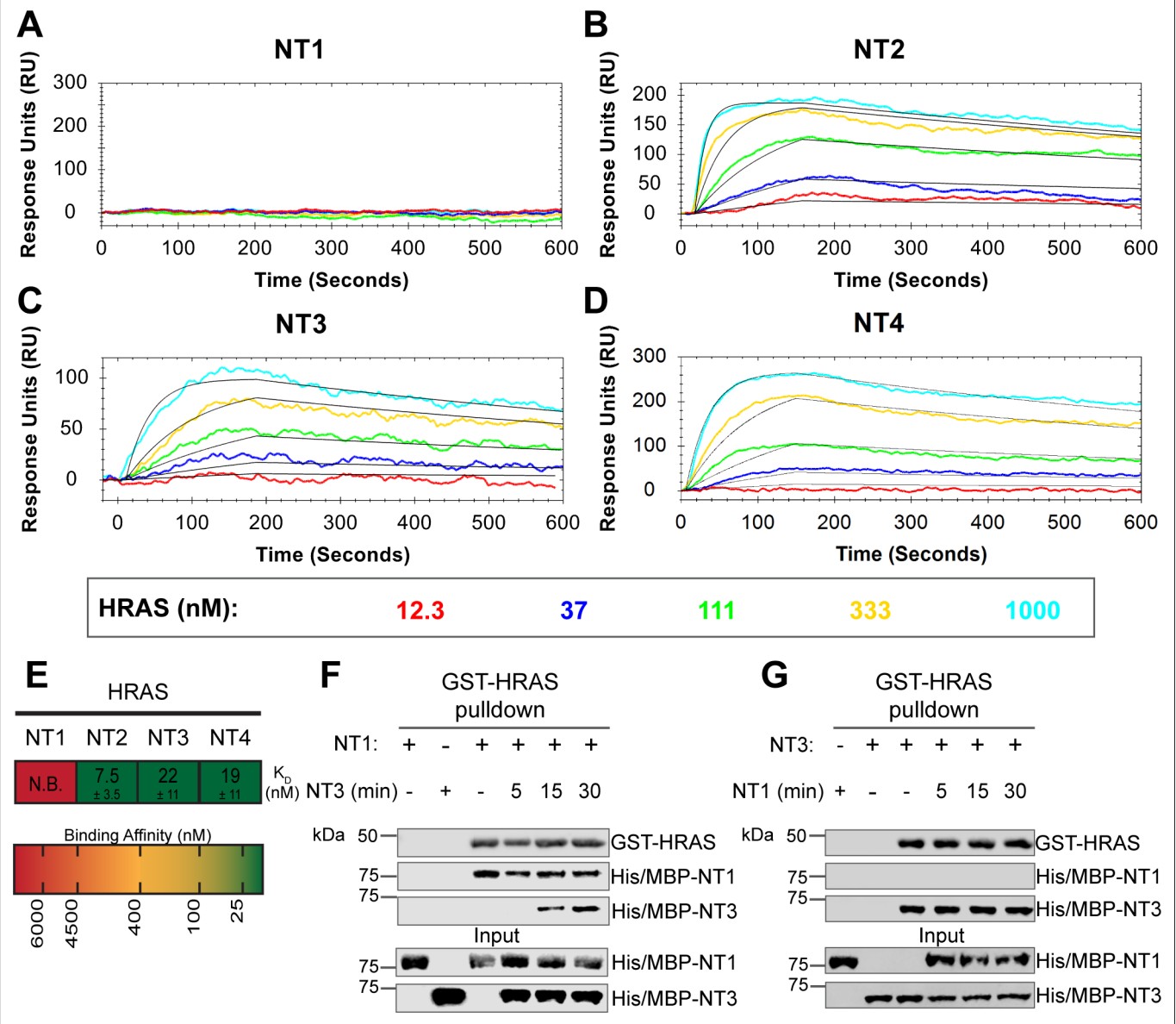

**Figure 3.** BRAF specific region (BSR) in conjunction with the cysteine rich domain (CRD) reduces binding affinity for HRAS. (**A–D**) Open surface plasmon resonance (OpenSPR) binding curves of HRAS flowed over NT1–4 immobilized on NTA sensors and the best fit curves (black) produced from a 1:1 fitting model kinetic evaluation. Representative of independent experiments with similar results each (NT1: n=7, NT2: n=2, NT3: n=2, NT4: n=3). (**E**) Diagram of the mean dissociation constant ($K_D$) ± standard deviation from independent OpenSPR experiments of HRAS flowed over immobilized NT1–4. (**F–G**) Western blot of purified His/MBP-NT1 (**F**) or -NT3 (**G**) binding to active GST-HRAS on glutathione resin in a pulldown assay and subsequent competition with NT3 (**F**) or NT1 (**G**). Representative of two independent experiments each with similar results.

The online version of this article includes the following source data and figure supplement(s) for figure 3:

**Source data 1.** (Panels a–e) Kinetic evaluation of 1:1 fits made in TraceDrawer for fitted curves NT2, NT3, and NT4.

**Source data 2.** (Panel f) Western blot of purified His/MBP-NT1 binding to active GST-HRAS on glutathione resin in a pulldown assay and subsequent competition with NT3.

**Source data 3.** (Panel g) Western blot of purified His/MBP-NT3 binding to active GST-HRAS on glutathione resin in a pulldown assay and subsequent competition with NT1.

**Figure supplement 1.** HRAS-NT1 open surface plasmon resonance (OpenSPR) shows slow association.

**Figure supplement 1—source data 1.** Western blot of GST-HRAS on glutathione resin and NT1 binding through pulldown assay.

show that NT1 binds minimally within 5 min and reaches maximal binding by 30 min (**Figure 3—figure supplement 1B**). These pulldown results are consistent with OpenSPR data and support that NT3 has a much faster and stronger association with HRAS and that NT1 is not able to outcompete NT3 for binding to HRAS.

## BSR differentiates the BRAF-KRAS interaction from the BRAF-HRAS interaction

To validate our hypothesis that the BSR negatively regulates BRAF activation in a RAS isoform-specific manner, we investigated the interactions between BRAF and KRAS. We purified GST-tagged, dimeric full-length KRAS4b (herein referred to as KRAS; **Figure 4—figure supplement 1**), which includes the C-terminal hypervariable region (HVR), in both active GMPPNP-loaded and inactive GDP-loaded forms. The HVR is an important region for regulating RAS isoform differences, like membrane anchoring, localization, RAS dimerization, and RAF interactions (**Prior and Hancock, 2012**). Since the RAS G domain is highly conserved, any observed differences between RAS isoforms are most likely a direct result of HVR impacts. Likewise, Terrell and colleagues have shown that exchanging RAS isoform HVRs enables the isoform to behave like its counterpart (**Terrell et al., 2019**). Pulldowns showed that active KRAS binds to all BRAF NT constructs, whereas inactive KRAS binds much less (**Figure 4A**). Using OpenSPR, we observed that BRAF NT1 binds active KRAS with a $K_D$ of 265±7 nM (**Figure 4B and C**), as opposed to HRAS, in which no binding was observed (**Figure 3A**). Additionally, we observed an average $K_D$ of 31±5 nM between KRAS and BRAF NT2 (**Figure 4B and D**). Inactive KRAS does not bind to BRAF NT2 nor does GST alone (**Figure 4—figure supplement 2**), confirming GMPPNP loading of KRAS. Parallel experiments show that BRAF NT3 and NT4 have binding affinities for active KRAS of 96±24 nM and 53±22 nM, respectively (**Figure 4B and E–F**). KRAS binds BRAF NT2, NT3, and NT4 with similar $K_D$ values, suggesting that for any significant RAS-RAF binding differences, BSR, RBD, and CRD must be present together (NT1).

Full-length His/MBP-KRAS was also recombinantly expressed, and the tag was cleaved through a TEV protease reaction to produce an untagged KRAS (**Figure 4—figure supplement 3A and B**). Pulldowns verify that untagged, active KRAS interacts with all BRAF NTs and untagged, inactive KRAS has much lower affinity (**Figure 4—figure supplement 3C**). GMPPNP loading of untagged KRAS was confirmed by stronger binding of active KRAS-GMPPNP than inactive KRAS (**Figure 4—figure supplement 3C and F**). Furthermore, kinetic analysis through OpenSPR shows that active GST-KRAS and untagged KRAS bind to BRAF NT2 with similar $K_D$ values indicating that the tag does not affect native interactions (**Figure 4B and C**, **Figure 4—figure supplement 3D and E**).

Truncated BRAF N-terminal proteins purified from *E. coli* clearly delineate a binding preference between RAS isoforms determined by specific BRAF domains, however, whether this distinction is maintained in more physiological conditions with the whole, active protein remained in question. Full-length (FL) BRAF was purified from HEK293F mammalian cells to produce a catalytically active form incorporating post-translational modifications and scaffold proteins (**Figure 4—figure supplement 4A**), as described and verified in **Cope et al., 2018**. To investigate whether RAS isoform specificity is maintained, we compared the FL-BRAF binding kinetics of HRAS and KRAS through OpenSPR. KRAS bound to FL-BRAF with a high affinity of 101±72 nM ( mean ± standard deviation) interaction, whereas HRAS displayed no interaction with FL-BRAF proving a low affinity interaction at >3 μM (**Figure 4G and H**). Inactive KRAS-GDP has a much lower affinity than active KRAS for FL-BRAF, exemplified by the much lower response at equal concentrations (**Figure 4—figure supplement 4B**). The NT1:KRAS and FL-BRAF:KRAS interactions do not possess significantly different $K_D$ values (unpaired t test p>0.05), establishing that NT1, and therefore other BRAF fragments, is representative of the full-length protein. Additionally, these results indicate that the KD and CR2 of BRAF do not affect the binding kinetics of the N-terminal interactions with RAS.

In efforts to better define the BSR-mediated specificity with KRAS, we performed HDX-MS on BRAF NT2 with KRAS. As expected, slower deuterium exchange was observed at the RBD peptides from BRAF 174–188 (**Figure 4I**), indicating a high affinity interaction. Peptides from within the BSR, specifically the two alpha-helices from residues 49–64 and 82–99, do not display any changes in deuterium rate exchange (**Figure 4I**). Unlike HRAS, KRAS binding does not perturb the structural rigidity of the BSR in a sizable manner (**Figure 2A**). All the time-dependent deuterium uptake plots for NT2 apo- and KRAS-bound are included in **Supplementary file 1**. Taken together, the BSR promotes differentiation

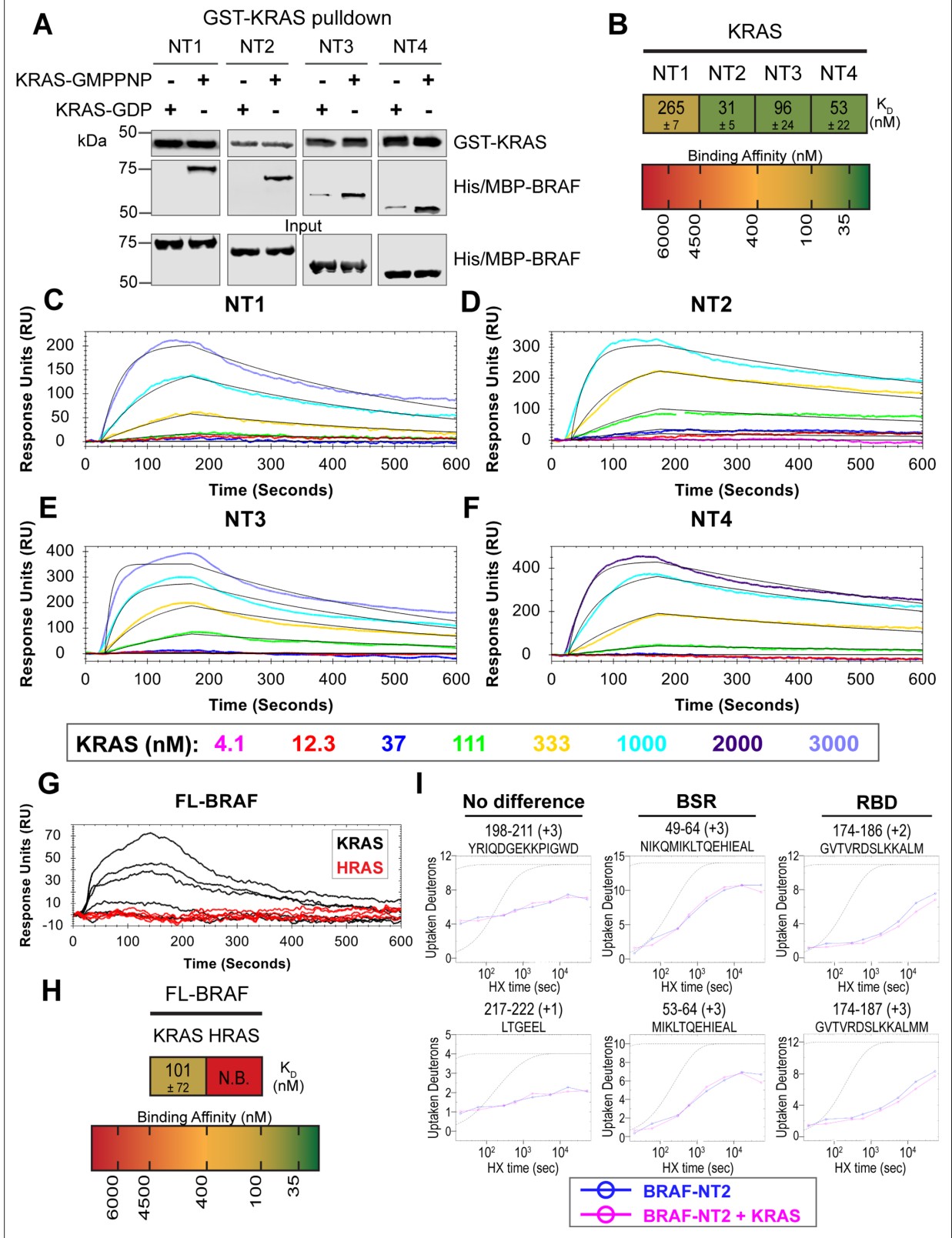

**Figure 4.** BRAF specific region (BSR) differentiates the BRAF-KRAS interaction from the BRAF-HRAS interaction. (**A**) Western blot of purified His/MBP-NT1–4 binding to GST-KRAS on glutathione resin in a pulldown assay. Representative of two independent experiments with similar results. (**B**) Diagram of the mean dissociation constant ($K_D$) ± standard deviation from independent open surface plasmon resonance (OpenSPR) experiments of KRAS flowed over immobilized NT1–4. (**C–F**) OpenSPR binding curves of KRAS flowed over NT1–4 immobilized on NTA sensors and the best fit curves (black)

*Figure 4 continued on next page*

*Figure 4 continued*

produced from a 1:1 fitting model kinetic evaluation. Representative of at least two independent experiments with similar results each. (**G**) OpenSPR binding curves of KRAS and HRAS flowed over FL-BRAF immobilized on NTA sensors and the best fit curves (black) produced from a 1:1 fitting model kinetic evaluation. K- and HRAS flowed over at increasing concentrations of 12.3, 37, 111, 333, 1000, and 3000 nM at 30 μL/min. Representative of two independent experiments with similar results each. (**H**) Diagram of the mean dissociation constant ($K_D$) ± standard deviation from independent OpenSPR experiments of H/KRAS flowed over immobilized FL-BRAF. FL-BRAF:KRAS compared to NT:KRAS unpaired t test p=0.0551. (**I**) Representative NT2 peptides identified from hydrogen-deuterium exchange mass spectrometry (HDX-MS) in the absence (blue) and presence (pink) of KRAS. Peptides with no difference in H-D exchange rate are exemplified under 'no difference'. Presented peptide plots displaying differences in H-D exchange rate are representative of a trend of 4+ overlapping peptides in the RAS binding domain (RBD). Gray dotted lines represent the theoretical exchange behavior for specified peptide that is fully unstructured (top) or for specified peptide with a uniform protection factor (fraction of time the residue is involved in protecting the H-bond) of 100 (lower).

The online version of this article includes the following source data and figure supplement(s) for figure 4:

**Source data 1.** (Panel a) Western blot of purified His/MBP-NT1-4 binding to GST-KRAS on glutathione resin in a pulldown assay.

**Source data 2.** (Panels b–f) Kinetic evaluations of 1:1 fits made in TraceDrawer for fitted curves NT1, NT2, NT3, and NT4.

**Source data 3.** (Panels g and h) Kinetic evaluations of 1:1 fits made in TraceDrawer for fitted curves full-length (FL) BRAF.

**Source data 4.** (Panel i) Raw data used to plot curves with ExMS2.

**Figure supplement 1.** SEC of active GST-HRAS and GST-KRAS.

**Figure supplement 1—source data 1.** Coomassie stained gel of GST-KRAS final purification product.

**Figure supplement 2.** Active GST-KRAS specifically binds to BRAF-NTs.

**Figure supplement 2—source data 1.** Coomassie stained gel of GST protein, purified following the same protocol as GST-HRAS.

**Figure supplement 3.** Characterizing untagged KRAS.

**Figure supplement 3—source data 1.** (Panel c) Western blot of purified KRAS binding to His/MBP-NT1–4 on amylose resin in a pulldown assay.

**Figure supplement 3—source data 2.** (Panels d–f) Kietic evaluations of 1:1 fits made in TraceDrawer for fitted curves NT2.

**Figure supplement 4.** Full-length BRAF binds specifically to active GST-KRAS.

**Figure supplement 4—source data 1.** Coomassie stained gel of purified recombinant full-length (FL) BRAF with copurified chaperone proteins.

between interactions with H- and KRAS isoforms and moderates a preference for KRAS by allowing fast and strong association when the entire N-terminal region is present as opposed to stalling the interaction with HRAS.

## BSR and CRD promote BRAF autoinhibitory interactions

In addition to interacting with RAS, the BRAF N-terminal regulatory region is also important in maintaining the autoinhibited conformation (*Cutler et al., 1998*; *Winkler et al., 1998*; *Tran et al., 2005*). Specifically, structures of autoinhibited BRAF in complex with MEK and 14-3-3 reveal that the CRD makes key interactions with the KD (*Park et al., 2019*; *Martinez Fiesco et al., 2022*). The BSR is not resolved in this structure, however, and much less is known about the roles of BSR and RBD in BRAF autoinhibition. To better understand the intra-domain interactions involved in BRAF autoinhibition, we investigated the binding preferences of the four NT (NT1–4) constructs to the N-terminally 6xHis-tagged BRAF-KD purified from *E. coli*. We performed pulldown experiments with BRAF constructs NT1–4, in which biotinylated BRAF-KD was captured on streptavidin beads and probed for bound His/MBP-tagged BRAF NTs. Analysis through western blotting showed that NT1 and NT3 do indeed bind to KD, but NT2 and NT4 do not bind (*Figure 5A*). These results show that the CRD is necessary and that BSR and RBD are not the primary contacts in autoinhibitory interactions with KD.

These results were further validated through OpenSPR experiments, which quantified the first in vitro binding affinity values for BRAF autoinhibition interactions. By immobilizing KD on a carboxyl sensor and flowing over increasing concentrations of NT1 (5, 15, 44, 133, 400 nM) at a flow rate of 30 μL/min, we observed specific binding between KD and NT1, with a $K_D$ of 11±1.5 nM (*Figure 5B and C*). Similarly, we removed the His/MBP-tag from BRAF NT1 through a TEV protease cleavage reaction and flowed over untagged NT1. Kinetic analysis confirmed that the interaction is preserved with the $K_D$ = 13 nM (*Figure 5—figure supplement 1*). Parallel OpenSPR experiments with KD and NT3 produced an average $K_D$ of 54±24 nM, higher than the dissociation constant between KD and NT1 (unpaired t test p<0.05; *Figure 5B and D*). The difference between NT1 and NT3 is the inclusion and exclusion of the BSR, respectively. This shows that the presence of the BSR increases the

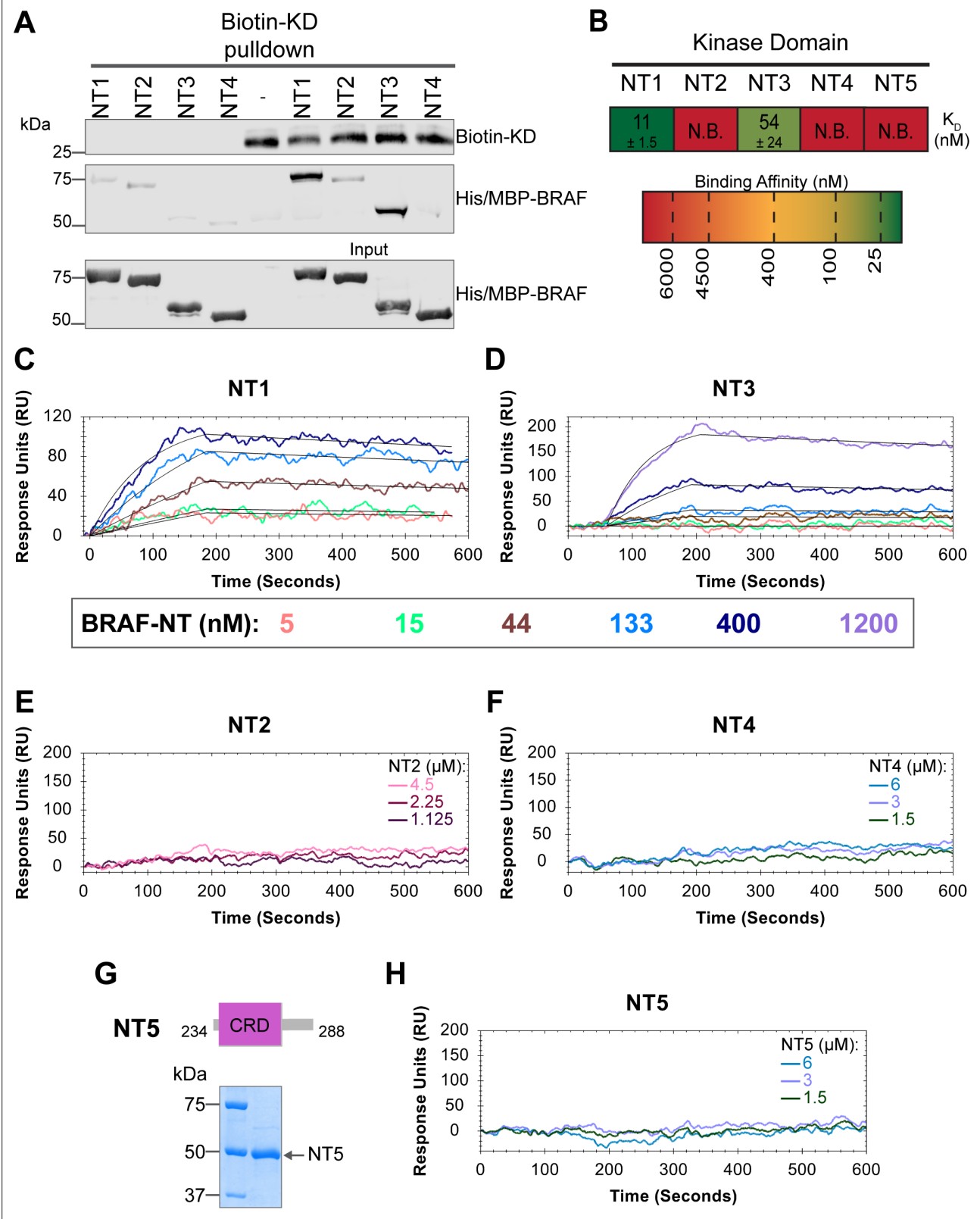

**Figure 5.** BRAF specific region (BSR) and cysteine rich domain (CRD) promote BRAF autoinhibitory interactions. (**A**) Western blot of purified His/MBP-NT1–4 binding to biotinylated His-KD on streptavidin beads in a pulldown assay. Representative of three independent experiments with similar results. (**B**) Diagram of the mean dissociation constant ($K_D$) ± standard deviation from independent open surface plasmon resonance (OpenSPR) experiments of NTs flowed over immobilized KD. $K_D$ of NT1 compared to NT3 unpaired t test p=0.0423. (**C–D**) OpenSPR binding curves of NT1 and NT3 at 5, 15, 44,

*Figure 5 continued on next page*

*Figure 5 continued*

133, 400, and 1200 nM (NT3 only) flowed over KD and the best fit curves (black) produced from a 1:1 fitting model kinetic evaluation. Representative of independent experiments with similar results (NT3: n=*2*, NT1: n=*3*). (**E–F**) No binding of NT2 or NT4 to immobilized KD was observed by OpenSPR even at high concentrations (NT2: 1.125, 2.25, 4.5 µM; NT4: 1.5, 3, 6 µM). Representative of two independent experiments each with similar results. (**G**) Diagram of BRAF NT5 (top) and Coomassie stained gel of recombinant NT5 (lower). (**H**) NT5 at 1.5, 3, and 6 µM flowed over immobilized KD on a carboxyl sensor. Representative of three independent experiments.

The online version of this article includes the following source data and figure supplement(s) for figure 5:

**Source data 1.** (Panel a) Western blot of purified His/MBP-NT1–4 binding to biotinylated His-KD on streptavidin beads in a pulldown assay.

**Source data 2.** (Panels b–f, h) Kinetic evaluations of 1:1 fits made in TraceDrawer for fitted curves NT1 and NT3.

**Source data 3.** (Panel g) Coomassie stained gel of recombinant NT5.

**Figure supplement 1.** MBP tag does not interfere with the interaction between NT1 and KD.

affinity for KD, further implicating that the BSR is important for stabilizing the autoinhibited state. The OpenSPR experiments with NT2 and NT4 showed very little specific binding even at higher concentrations of analyte (NT2: 1.125, 2.25, 4.5 µM; NT4: 1.5, 3, 6 µM; *Figure 5E and D*) with $K_D$ values >4.5 µM (NT2) and >6 µM (NT4).

Although the CRD is the main contact point for BRAF autoinhibitory interactions, whether this domain is sufficient to maintain the interactions remained in question. Tran et al. previously showed that the BRAF autoinhibitory domain is minimally constrained to residues 100–345 (RBD+CRD; *Tran et al., 2005*), yet recent cryo-EM structures do not detect any interactions with the KD outside of the CRD (*Park et al., 2019*; *Martinez Fiesco et al., 2022*). Therefore, we designed and purified a BRAF-CRD only protein (herein referred to as NT5; residues 234–288; *Figure 5G*) to test the binding affinity of this domain for KD. No binding of NT5 and KD was detected through OpenSPR (*Figure 5B and H*), revealing that the CRD alone is not adequate to establish interactions with the KD. A low affinity $K_D$ may occur at >6 µM, however, together our data indicates that the RBD and CRD are essential for high affinity interactions with the KD and the BSR increases the strength of the interactions.

## The RAS-RAF interaction directly disrupts RAF autoinhibition

Given the implication of RAS in relieving RAF autoinhibition, we performed pulldown experiments with HRAS, KD, and NT1 to investigate how RAS affects the association of the N- and C-terminal domains of BRAF. Specifically, GST-HRAS was captured on glutathione resin and probed for interactions between NT1 and KD. In this experiment, equal molar ratios of NT1 and KD were pre-incubated for 1 hr to allow for binding, mimicking the autoinhibited monomeric BRAF. Then HRAS-bound resin was added, representing the activation event following RAS association. By performing the pulldown in this manner, we capture the specific population of NT1 that is bound to HRAS. Two populations of NT1 are possible: one bound to HRAS and the other bound to KD. Therefore, the experimental design of pulldowns is important to distinguish between populations and accurately depict the protein-protein interactions. The pulldowns clearly showed that NT1 was associated with HRAS on the resin, while no KD was found in the fraction of NT1 that was bound to HRAS. These results demonstrate that KD is unable to bind to NT1 in the presence of HRAS (*Figure 6A*).

Our pulldown experiments were complemented with OpenSPR experiments. NT1 and HRAS were pre-incubated in equal molar amounts (400 nM for each) for 1 hr, allowing for sufficient binding prior to injecting onto the KD-immobilized sensor. A comparison between binding of NT1 and KD with and without HRAS showed that the presence of HRAS abolished the interaction between NT1 and KD (*Figure 6B*). These findings demonstrate that HRAS binding to BRAF directly relieves BRAF autoinhibition by disrupting the NT1-KD interaction, providing in vitro evidence of RAS-mediated relief of RAF autoinhibition, the central dogma of RAS-RAF regulation.

## Oncogenic BRAF-KD[D594G] has decreased affinity for NT1 and thus attenuates autoinhibitory interactions

Although oncogenic mutations in BRAF are thought to relieve autoinhibitory interactions thereby promoting dimerization and activation, there is currently no direct evidence to support this hypothesis. To investigate this further, we purified the 33 kDa BRAF-KD with oncogenic mutation D594G (referred to as KD[D594G]). BRAF[D594G] is the most common BRAF mutant in non-small cell lung cancer patients and

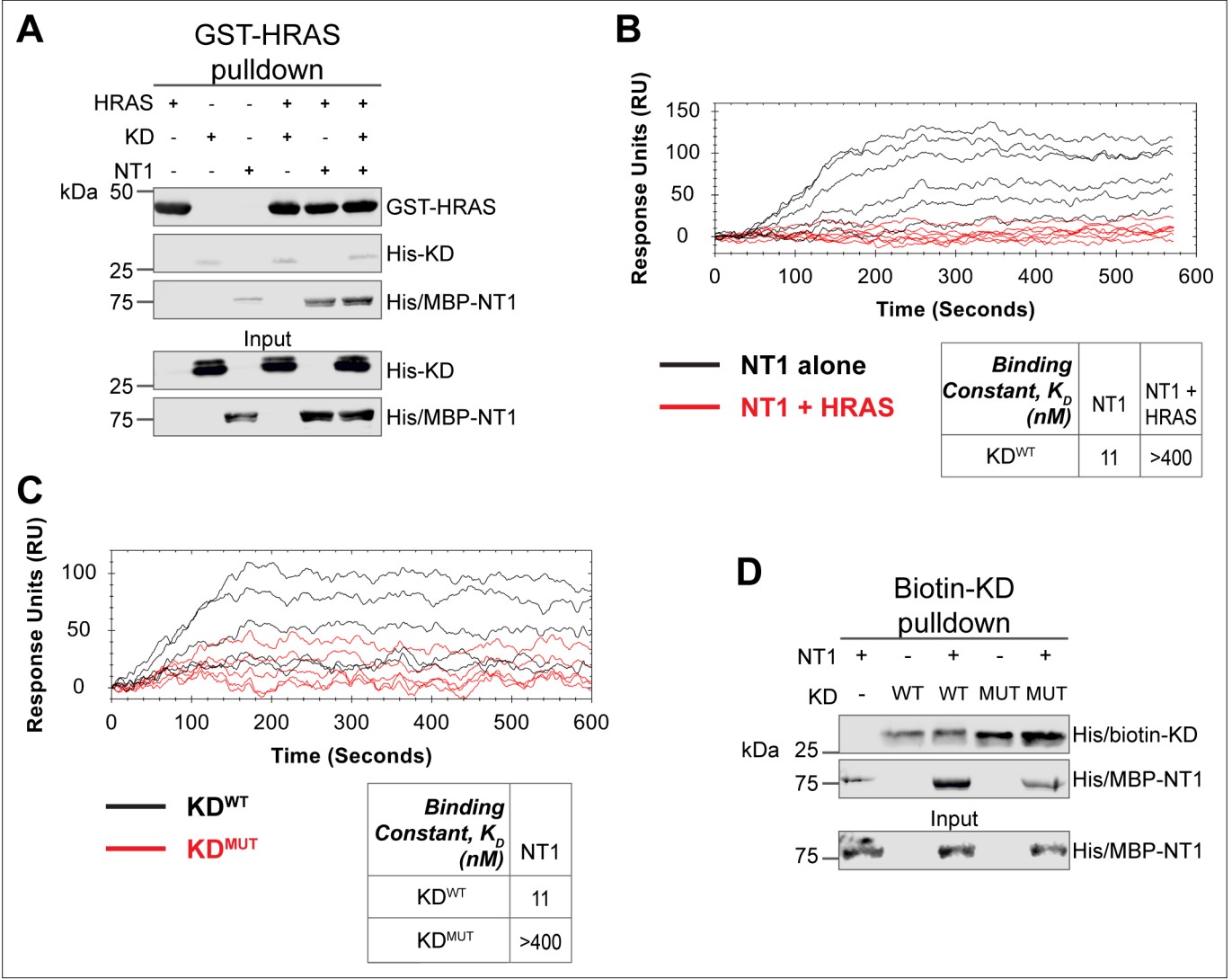

**Figure 6.** HRAS and KD^D594G disrupt BRAF autoinhibition. (**A**) Western blot of pulldown assay of pre-incubated His/MBP-NT1 and His-KD added to purified active GST-HRAS on glutathione resin. Representative data of two independent biological replicates with similar results. (**B**) Open surface plasmon resonance (OpenSPR) experiments in which NT1 at 5, 15, 44, 133, 400 nM (black) and NT1 + HRAS-GMPPNP (1:1) at 5, 15, 44, 133, and 400 nM (red) flowed over KD immobilized on carboxyl sensors. Representative data of three independent biological replicates with similar results. (**C**) OpenSPR experiments of NT1 at 5, 15, 44, 133, and 400 nM flowed over immobilized KD^WT (black) or KD^MUT (D594G; red) on carboxyl sensors. Representative data of three independent biological replicates with similar results. (**D**) Western blot of purified His/MBP-NT1 binding to either biotinylated His-KD^WT or His-KD^MUT (D594G) in pulldown assays on streptavidin beads. Representative data of three independent biological replicates with similar results.

The online version of this article includes the following source data for figure 6:

**Source data 1.** (Panel a) Western blot of pulldown assay of pre-incubated His/MBP-NT1 and His-KD added to purified active GST-HRAS on glutathione resin.

**Source data 2.** (Panel d) Western blot of purified His/MBP-NT1 binding to either biotinylated His-KD^WT or His-KD^MUT (D594G) in pulldown assays on streptavidin beads.

has been identified as oncogenic despite its completely dead kinase activity (class 3 BRAF mutant; *Nieto et al., 2017*). Previous studies have shown that the D594G mutant has higher dimerization potential than wild-type BRAF (*Cope et al., 2020*), but it remains unclear how this mutation relieves the dimer interface from autoinhibitory interactions. To test our hypothesis that KD^D594G has lower autoinhibitory potential, we analyzed the binding affinity of NT1 and KD^D594G through OpenSPR and pulldown assays. We immobilized KD^D594G on a carboxyl sensor to the same response level as KD^WT (~6000 RUs) and flowed over the same concentrations of NT1 in both sets of experiments. With

$KD^{D594G}$, we observed less binding to NT1 compared to $KD^{WT}$, and the responses were too low to calculate a $K_D$ (**Figure 6C**). These results were validated using a pulldown experiment in which both KD proteins were biotinylated and pulled down on streptavidin beads. Binding of NT1 was subsequently probed through western blot (**Figure 6D**) and little to no NT1 was bound to $KD^{D594G}$, in direct contrast to $KD^{WT}$. These findings support our hypothesis that the D594G mutation significantly decreases the autoinhibitory interactions, rendering its oncogenic potential.

## Discussion

In this study, we investigated the regulation of BRAF by examining the interactions between its N-terminal domain and RAS, as well as its C-terminal catalytic domain. The BRAF activation mechanism involves a nuanced set of events that is regulated by individual functions of the BRAF N-terminal domains in concert with the BRAF catalytic domain and RAS binding. Our findings, which were obtained from pulldowns, OpenSPR, and HDX-MS, suggest that the BSR in conjunction with the RBD and CRD impedes rapid HRAS binding, while permitting interaction with KRAS. Additionally, our

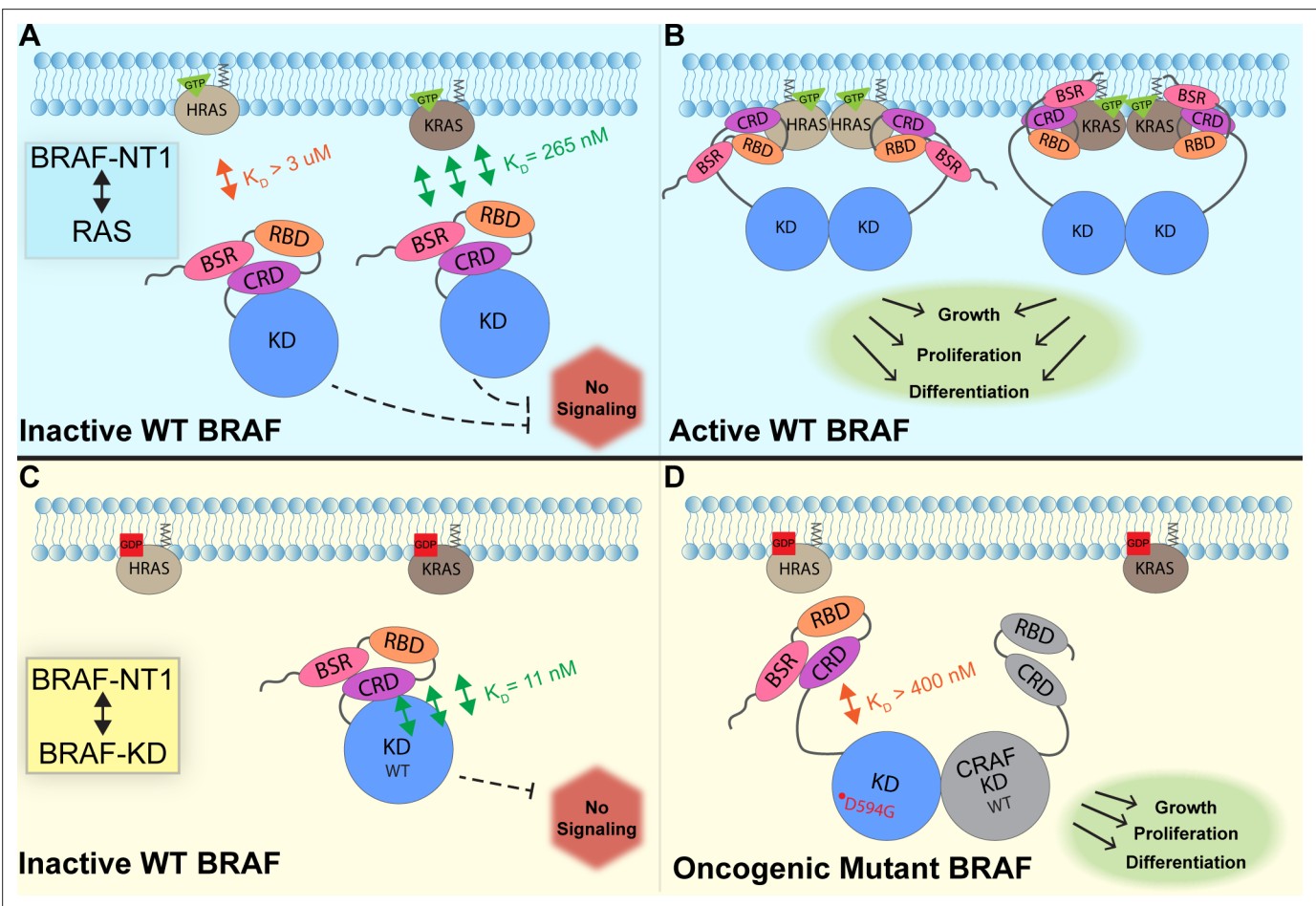

**Figure 7.** Model of BRAF activation. (**A**) BRAF is initially an autoinhibited monomer in the cytosol, in which signaling through the RAS-RAF-MEK-ERK cascade is not promoted. The BRAF N-terminal region (NT1; amino acids 1–288), including the BRAF specific region (BSR), cysteine rich domain (CRD), and RAS binding domain (RBD), interacts with active GTP-bound RAS at the membrane in an isoform-specific manner (relative positions of N-terminal domains may not reflect all interactions). BRAF has higher affinity for KRAS due to isoform differences, as shown through the different dissociation constants ($K_D$) determined through open surface plasmon resonance (OpenSPR). (**B**) Once bound to active H- or KRAS, BRAF is unable to remain in the autoinhibited conformation and is subsequently activated upon dimerization, which stimulates signaling for events such as cell growth, proliferation, and differentiation. (**C**) Tight binding is observed with the BRAF kinase domain (KD; amino acids 442–723) and BRAF NT1, revealing the concerted action of the BSR, RBD, and CRD domains to reinforce the autoinhibited conformation and restrict signaling without upstream activation. (**D**) Oncogenic $BRAF^{D594G}$ stimulates activation of MAPK pathway through a decreased ability to remain in the autoinhibited conformation and an increased potential to dimerize with CRAF. Evading autoinhibitory regulation leads to overactivation of the signaling cascade and tumorigenesis.

investigation reveals that the BRAF N-terminal region has high affinity for BRAF-KD, providing further direct evidence that the N-terminal domains facilitate an autoinhibitory conformation. In contrast, mutations such as D594G in BRAF-KD can abrogate the tight autoinhibitory interactions. We propose a mechanism for BRAF regulation: the BSR works in concert with the CRD to foster RAF isoform-specific activity by favoring association with KRAS over HRAS. Simultaneously, BSR and CRD joined by RBD play a pivotal role in maintaining autoinhibition by tightly interacting with the KD. Binding of RAS to the N-terminal region, primarily mediated by RBD, directly relieves BRAF autoinhibition, ultimately leading to activation. In the context of oncogenic BRAF, mutations in the catalytic domain can also relieve BRAF autoinhibition and aberrantly upregulate MAPK signaling (*Figure 7*).

RAS oncogenes are prominent drivers of human tumors, acting through effectors such as RAF and initiating hyper-activation of the MAPK pathway. The interaction between the RBD-CRD of RAF and the switch I and interswitch region of RAS has previously been established (*Tran et al., 2021*), but the molecular mechanism behind RAF activation upon RAS binding is not fully understood. While a number of structural studies have recently emerged, a comprehensive analysis of the effects of RAS binding on BRAF relief of autoinhibition and activation is needed to design new therapies. Our in vitro analyses of the individual binding profiles of distinct regulatory regions of BRAF add to the growing body of evidence about BRAF regulation and activation. We isolated the three regulatory domains in BRAF to show their direct effects on RAS binding through quantitative $K_D$ values. Since NT2, NT3, and NT4 have similar binding affinities for HRAS and KRAS, our findings support that the RBD is the primary driver of the RAS-RAF interaction, as expected. The presence of the BSR and CRD in NT1 clearly slows the ability for HRAS, but not KRAS, to bind, revealing a critical regulatory role for this region. This evidence is further supported by FL-BRAF, which has high affinity for KRAS but not HRAS. SPR experiments from Tran et al. that examined the KRAS-CRAF interaction found that the presence of the CRD increases the binding affinity to KRAS significantly (*Tran et al., 2021*). The authors conclude that the CRD is important for differentiating between the RAS GTPase superfamily, even though high sequence homology exists between the switch I region and the RBDs of RAF (*Tran et al., 2021*). Interestingly, our results show that the CRD alone does not affect the affinity of BRAF toward HRAS or KRAS, with binding affinity values within the same nanomolar range for constructs NT2–4. This could be due to RAF isoform differences, as slight amino acid differences are found in the sequence despite the conservation of the CRD in RAF kinases. Certain CRAF-CRD mutations that are key CRD-KRAS contacts were shown to increase the $K_D$ values of binding to KRAS (*Tran et al., 2021*). The mutation with the most prominent difference, K179, corresponds with sequence divergence between B- and CRAF. While KRAS and HRAS are identical in the interswitch region where the CRD interacts, RAF isoform differences may provide an additional layer of regulation in RAS-mediated activation.

RAS isoforms are believed to have distinct impacts on effector binding and activation. Although these isoforms are highly conserved, the C-terminal HVR distinguishes HRAS from KRAS in significant ways. Cell-based assays suggest that KRAS recruits and activates CRAF more efficiently than HRAS (*Yan et al., 1998*), whereas in vitro binding studies found that unmodified HRAS associates with BRAF but not CRAF (*Fischer et al., 2007*). However, BRET assays suggest that CRAF does not show preference for either H- or KRAS, while BRAF appears to prefer KRAS (*Terrell et al., 2019*). This preference is suggested to result from the potential favorable interactions between the negatively charged BSR of BRAF and the positively charged, poly-lysine region of the HVR of KRAS (*Terrell et al., 2019*). The conflicting studies highlight the complexity of this signaling pathway. Our binding data not only reaffirm the presence of isoform-specific activity but also offer insight into the underlying mechanisms. We speculate that the diminished BRAF NT1 binding to HRAS coupled with increased flexibility observed in the BSR upon HRAS binding as demonstrated through HDX-MS, may be due to an inherent incompatibility between HRAS-HVR and the BSR as well as strong intramolecular forces within the regulatory region, which initially impedes a rapid interaction between HRAS and the N-terminal domains. Our pulldown experiments and OpenSPR analyses confirm the isoform-specific preference of NT1 and FL-BRAF for KRAS. Terrell et al. propose that molecular interactions and isoform-specific binding preferences arise from charge attraction between the BSR and KRAS-HVR (*Terrell et al., 2019*). While our results do not directly show binding interactions between KRAS to the BSR through slowed H-D exchange, they do not rule out the possibility of a transient, low-affinity interaction or proximity between the KRAS-HVR and BRAF-BSR. Notably, the absence of slowed H-D exchange suggests that the KRAS-HVR does not disturb the conformational stability of the BSR and is not inherently

incompatible with the BSR. Our data combined with findings by Terrell et al. imply that the KRAS-HVR may enhance the favorability of KRAS interactions with the RBD through allosteric modulation of the BSR. Our results provide insight into the complex isoform-specific interactions, which are clearly important given that RAS and RAF isoforms display specific levels of activity in mutant and wild-type cells. For example, KRAS is responsible for most RAS-driven cancers (*Simanshu and Morrison, 2022*), and BRAF has the highest basal activity among RAF isoforms (*Emuss et al., 2005*).

In recent years, two significant studies resolved the structure of autoinhibited, monomeric BRAF in complex with MEK and 14-3-3 (*Park et al., 2019*; *Martinez Fiesco et al., 2022*). These studies revealed that the CRD is cradled within 14-3-3 protomers, which interact with phosphoserines at the N- and C-terminal regions of BRAF. In this conformation, the CRD interacts with the KD, while the membrane binding loops of the CRD are occluded by 14-3-3 (*Park et al., 2019*). Our results showing that NTs without the CRD do not bind to the KD support the critical role of the CRD as necessary for autoinhibitory interactions, yet insufficient alone. Because the BSR is believed to be highly flexible, it was not resolved in either cryo-EM structure. Comparison of constructs with and without the BSR shows that the BSR, in synchronism with the CRD, increases the affinity of BRAF for the KD, thereby revealing that the BSR could be a promising candidate for further study as a potential target. Targeting the BSR could reinforce the autoinhibitory conformation, thereby preventing BRAF activation by blocking RAS interaction. Interestingly, our results indicate that NT1 has a higher affinity for KD than NT3, aligning well with the previous report that BRAF has higher autoinhibitory activity than CRAF (*Spencer-Smith et al., 2022*), as NT3 of BRAF is more alike to the N-terminal of CRAF which does not have the BSR. The reported differences between BRAF-CRD and CRAF-CRD by *Spencer-Smith et al., 2022*, may not only stem from the CRD isoform differences, but also from the presence of the BSR in BRAF providing a stabilizing autoinhibitory effect. We speculate that the BSR in BRAF exhibits an allosteric effect to enforce the autoinhibitory interactions between the CRD and KD. The CRD is a hotspot of RASopathy mutations in BRAF, with the most common mutant, Q257R, activating the MAPK pathway (*Niihori et al., 2006*; *Rodriguez-Viciana et al., 2006*). While RASopathy mutations in BRAF are centered within the CRD, our results imply that the mutations may also affect the ability of the BSR to regulate BRAF autoinhibition.

Recent studies have partially revealed the order of RAF activation events through static structures (*Park et al., 2019*; *Tran et al., 2021*; *Cookis and Mattos, 2021*; *Martinez Fiesco et al., 2022*) and live-cell studies with mutant KRAS (*Spencer-Smith et al., 2022*). Martinez Fiesco et al. resolved the RBD in the autoinhibited BRAF-MEK-14-3-3 complex, which revealed that the RBD forms a large interface with one of the 14-3-3 protomers and that RBD-KRAS contact residues are exposed and available to form bonds with RAS (*Martinez Fiesco et al., 2022*). Structural analysis predicts that the RBD-CRD interactions with RAS and the membrane disrupt the autoinhibitory interactions with 14-3-3 and the KD due to overlap in these binding regions (*Simanshu and Morrison, 2022*). We show, through definitive biochemical methods, that RAS binding directly forces BRAF out of autoinhibition and primes it for subsequent activating steps. Our in vitro binding studies align with previous implications that RAS relieves RAF autoinhibition shown through cell-based coIPs (*Tran et al., 2005*). Additionally, we found that the oncogenic potential of BRAF[D594G] is propelled through a decreased autoinhibitory potential. Together, these findings suggest that the autoinhibited conformation of wild-type BRAF is not capable of dimerizing, further supporting the model that BRAF binding to HRAS relieves autoinhibition and initiates the activation process. Once autoinhibition is relieved, subsequent dimerization becomes possible, leading to full activation of BRAF. Class 3 BRAF mutants, like BRAF[D594G], have been shown to bind to active RAS more efficiently than BRAF[WT] (*Yao et al., 2017*). This may be due to attenuated autoinhibitory interactions that make the RBD and CRD more accessible to RAS. These observations demonstrate the importance of considering mutations from an autoinhibitory standpoint and therapeutically maintaining autoinhibition to treat RAS-dependent BRAF cancers.

We present a comprehensive set of in vitro quantitative binding affinities for BRAF NTs, BRAF-KD, HRAS, and KRAS, shedding light on the roles of BRAF N-terminal domains in activation and regulation. Our in vitro studies were conducted using proteins purified from *E. coli*, which lack post-translational modifications and without involvements from cellular components such as the membrane and regulatory, scaffolding, or chaperone proteins that are involved in BRAF regulation. Nonetheless, our study provides a direct characterization of the intra- and inter-molecular protein-protein interactions involved in BRAF regulation, without the complications and false positives that can arise in cell-based

assays, which often cannot distinguish between mere proximity and biochemical interactions. Binding kinetic analysis showed that FL-BRAF purified from mammalian cells interacts with KRAS with the same affinity as NT1, providing strong evidence that the truncated BRAF constructs used in our study can represent FL-BRAF and the physiological conditions with post-translational modifications, native intramolecular interactions, and potential conformational changes that occur upon binding. Furthermore, the protein-protein interactions described here align with multiple literature sources that use cell systems or purified proteins from insect or mammalian cell lines (*Park et al., 2019*; *Fischer et al., 2007*; *Tran et al., 2021*; *Tran et al., 2005*). Our findings extend current knowledge of BRAF regulation and provide novel insight into the development of next-generation BRAF therapy. Our study reveals the previously understudied BSR as a critical modulator of BRAF activation and RAS binding specificity. Additionally, our experimental setup provides a potential framework for testing inhibitors against RAS-RAF interaction. Allosteric inhibitors stabilizing the autoinhibited conformation of BRAF could be a promising strategy against oncogenic KRAS, especially since it is the primary RAS isoform responsible for activating BRAF.

# Materials and methods

## Plasmids

GST-HRAS and His/MBP-KRAS were purchased from Addgene (#55653 and # 159546, respectively). GST-KRAS was created with standard Gibson Assembly (NEB) procedures with pGEX2T as the vector and the following primers: 5'ATCTGGTTCCGCGTGGATCCACTGAATATAAACTTGTGGTAG 3' (GST-KRAS_For), 5' CAGTCAGTCACGATGAATTCTTACATAATTACACACTTTGTCTTTG 3' (GST-KRAS_ Rev), 5' GAATTCATCGTGACTGACTGACG 3' (GST-vector_For), 5' GGATCCACGCGGAACCAG 3' (GST-vector_Rev). 6xHis-KD$^{WT}$ and 6xHis-KD$^{D594G}$ were designed as described previously (amino acids 442–723 with 16 solubilizing mutations: I543A, I544S, I551 K, Q562R, L588N, K630S, F667E, Y673S, A688R, L706S, Q709R, S713E, L716E, S720E, P722S, and K723G; *Cope et al., 2020*). His/MBP-BRAF NTs were created with standard Gibson Assembly (NEB) procedure in the pET28-MBP vector from the following primers: 5' CATATGCTCGGATCCGCGGCGCTGAGCGGTG 3' (BRAF-RBD_For 1), 5' CATATGCTCGGATCCTCACCCACAAAAACCTATCGTTAG 3' (BRAF-RBD_For 151), 5' GTTGTAAG AATTCAAGCTTACAACACTTCCACATGCAATTC 3' (BRAF-RBD_Rev 227), 5' CAAAGAACTGAATTCA AGCTTACAAATCAAGTTGGT 3' (BRAF-RBD_Rev 288). 5' agcaaatgggtcgcggatccACACACAACTTT GTACGAAAAAC'3 (BRAF-CRD_For), 5'cgagtgcggccgcaagcttaCAAATCAAGTTGGTCATAATTAAC'3 (BRAF-CRD_Rev).

## Protein expression

All protein construct plasmids were transformed into BL21 codon + *E. coli.* and grown to an OD$_{600}$ 0.6–0.8 in LB broth (BRAF NT constructs supplemented with 100 µM ZnCl$_2$), followed by induction with 0.4 mM IPTG. Cells were left overnight at 18°C, shaking 210 rpm. Cells were pelleted, flash-frozen, and stored at –80°C.

## GST-HRAS/KRAS purification

GST-tagged full-length HRAS/KRAS pellet was thawed and incubated with lysis buffer (20 mM HEPES pH 7.4, 150 mM NaCl, 1 mM EDTA, 5% glycerol, 1 mg/mL lysozyme, and protease inhibitor cocktail) for 1 hr at RT. Whole cell lysate was exposed to brief sonication (HRAS) or passed through the French Pressure Cell Press (KRAS) at 1250 psi and centrifuged. Soluble cell lysate was incubated with pre-equilibrated glutathione resin for 1 hr at 4°C. After extensive washing, HRAS/KRAS protein was eluted off the resin with elution buffer (20 mM HEPES pH 7.4, 150 mM NaCl, 1 mM EDTA, 5% glycerol, and 20 mM reduced glutathione). To dissociate bound nucleotide, HRAS/KRAS was incubated in HEPES buffer with 10 mM EDTA and 10 M excess GMPPNP (Sigma-Aldrich) for 30 min at 4°C. To allow rebinding, MgCl$_2$ was added to a final concentration of 20 mM and rotated for 2 hr at 4°C (*Amendola et al., 2019*). HRAS/KRAS was further purified on a Superdex 200 10/300 GL size exclusion chromatography column (Cytiva). Main steps were checked with SDS-PAGE followed by Coomassie staining. After concentration, aliquots were flash-frozen and stored at –80°C.

## Untagged KRAS purification

After overexpression and induction, cells were pelleted and resuspended in lysis buffer (50 mM HEPES pH 7.4, 300 mM NaCl, 1 mM TCEP, 5 mM MgCl$_2$, and protease inhibitor cocktail). Whole cell lysate

was passed through the French Pressure Cell Press (SLM-Aminco) twice at 1250 psi for lysis and centrifuged. Soluble cell lysate was incubated with pre-equilibrated TALON resin (TakaraBio) for 2 hr at 4°C. After extensive washing, KRAS was eluted off the resin with elution buffer (20 mM HEPES pH 7.4, 300 mM NaCl, 1 mM TCEP, 5 mM $MgCl_2$, and increasing imidazole concentrations [90, 500 mM]). To cleave the MBP-tag, TEV protease was added in 1:5 ratio to eluted KRAS and was dialyzed for 3 days at 4°C in buffer while reaction proceeded (20 mM HEPES pH 7.4, 300 mM NaCl, 1 mM TCEP, 5 mM $MgCl_2$). Solution was reapplied to TALON resin and supernatant containing KRAS was collected. To dissociate bound nucleotide, KRAS was incubated in HEPES buffer with 10 mM EDTA and 10 M excess GMPPNP (Sigma-Aldrich) for 30 min at 4°C. To allow rebinding, $MgCl_2$ was added to a final concentration of 20 mM and rotated for 2 hr at 4°C (*Amendola et al., 2019*). KRAS was further purified on a Superdex 75 10/300 GL size exclusion chromatography column (Cytiva). Main steps were checked with SDS-PAGE followed by Coomassie staining. After concentration, aliquots were flash-frozen and stored at –80°C.

## BRAF NT purification

After overexpression and induction, cells were pelleted and resuspended in lysis buffer (20 mM HEPES pH 8, 150 mM NaCl, 5% glycerol, and protease inhibitor cocktail). Whole cell lysate was exposed to brief sonication and centrifuged. Soluble cell lysate was incubated with pre-equilibrated Ni-NTA resin for 1 hr at 4°C. After extensive washing, BRAF NT protein was eluted off the resin with elution buffer (20 mM HEPES pH 7.4, 150 mM NaCl, 5% glycerol, and increasing imidazole concentrations [90, 200, 400 mM]). BRAF NT was further purified on a Superdex 200 10/300 GL size exclusion chromatography column (Cytiva). Main steps were checked with SDS-PAGE followed by Coomassie staining. After concentration, aliquots were flash-frozen and stored at –80°C.

## FL-BRAF purification

FL-BRAF was expressed in HEK293F cells followed by the protocol described by *Cope et al., 2018*. In brief, the cell pellet was resuspended in lysis buffer (150 nM NaCl, 20 mM HEPES, pH 7.4, 1 mM $Na_3VO_4$, 1 mM PMSF, 10% glycerol, and protease inhibitor). Whole cell lysate was exposed to sonication and centrifuged. After slow rotation with resin and an ATP wash to remove excess chaperone protein HSP70, BRAF was eluted with 1xFLAG peptide (20 mM HEPES, pH 7.4, 150 mM NaCl, 10% glycerol, and 200 µg/mL). FL-BRAF was further purified through SEC, from which appropriate fractions were concentrated and flash-frozen.

## WT BRAF-KD purification

His-tagged wild-type BRAF-KD pellet was lysed in buffer (50 mM HEPES pH 8.0, 150 mM NaCl, 5% glycerol, 10 mg/mL lysozyme, and protease inhibitor cocktail) for 1 hr at RT. Whole cell lysate was exposed to brief sonication and centrifuged. Soluble cell lysate was incubated with pre-equilibrated cobalt resin for 2 hr at 4°C. Resin was washed three times with low salt buffer (50 mM HEPES pH 8, 150 mM NaCl, and 5% glycerol), three times with high salt buffer (50 mM HEPES pH 8, 400 mM NaCl, and 5% glycerol), and three more times with wash buffer (50 mM HEPES pH 7.4, 150 mM NaCl, and 5% glycerol). Protein was eluted off the resin with elution buffers (50 mM HEPES pH 7.4, 150 mM NaCl, 5% glycerol, and varying imidazole concentrations [400 mM, 200 mM, 90 mM]) starting with the lowest imidazole concentration to the highest for a total of 10 elution fractions. WT BRAF-KD was further purified on a Superdex 75 10/300 GL size exclusion chromatography column (Cytiva). Main steps were checked with SDS-PAGE followed by Coomassie staining. After concentration, aliquots were flash-frozen and stored at –80°C.

## BRAF-KD-D594G purification

His-tagged BRAF-KD-D594G was purified as previously described (*Cope et al., 2020*). In brief, cell pellet overexpressing BRAF-KD-D594G was thawed and resuspended in lysis buffer (50 mM phosphate buffer pH 7.0, 250 mM NaCl, 20 mM imidazole, 10% glycerol, and EDTA-free protease inhibitor cocktail tablet). Whole cell lysate was exposed to brief sonication and centrifuged. Soluble cell lysate was incubated with pre-equilibrated Ni-NTA resin for 1 hr at 4°C. Protein-resin complex was washed 5× 20 mL with chaperone removal buffer (50 mM HEPES pH 7.4, 5 mM ATP, 50 mM KCl, 20 mM $MgCl_2$, 20 mM imidazole) and another 5× 20 mL washes with low salt buffer (50 mM HEPES pH 7.4, 250 mM

NaCl, 20 mM imidazole, 10% glycerol). BRAF was eluted with increasing concentrations of imidazole (50–400 mM), pooled, and concentrated. Concentrated BRAF was further refined on a Superdex 200 10/300 GL column (Cytiva). Protein was concentrated, aliquoted, flash-frozen, and stored at –80°C.

## Pulldowns

All proteins were added in 1:1 stoichiometric ratio and incubated together in binding buffer (50 mM HEPES pH 7.4, 150 mM NaCl, 5% glycerol, and 0.125 mg/mL BSA) with 20 µL of Glutathione Sepharose 4b resin (Cytiva), amylose resin (NEB), or Pierce streptavidin magnetic beads (Thermo Scientific). After extensive washing (50 mM HEPES pH 7.4, 500 mM NaCl, 5% glycerol, and 0.125 mg/mL BSA), 30 µL 4× loading dye was added to resin. Supernatants were loaded and protein analyzed through SDS-PAGE. After transfer to nitrocellulose membrane, GST-HRAS/-KRAS was probed with GST antibody (Santa Cruz Biotechnologies SC-138), KRAS with RAS antibody (Cell Signaling 67648S), and BRAF NTs/ BRAF-KD with His antibody (Sigma SAB5600227). Finally, western blots were imaged on Cytiva *Typhoon* imager.

## Biotinylation

One mg 'No-weigh Sulfo NHS biotin' (Thermo Scientific) was resuspended in water and immediately added to BRAF-KD in 50 molar excess. Reaction was performed in Thermo Scientific Slide-A-Lyzer MINI Dialysis Unit. Mixture was left at RT rocking for 30–60 min, then dialysis unit placed in 500 mL dialysis buffer and left at RT for 2 hr with gentle mixing.

## Hydrogen-deuterium exchange mass spectrometry

Protein samples (BRAF NT2 [60 µM], NT3 [26 µM], HRAS [44 µM], or KRAS [44 µM], protein stock in 20 mM HEPES pH 7.4, 150 mM NaCl, 5% glycerol) were exposed to deuterated buffer ($D_2O$ solution containing 20 mM HEPES pH 7.4, 150 mM NaCl, 5% glycerol) by mixing protein stock with $D_2O$ buffer in a 1:5 (vol:vol) ratio for times ranging from 20 s to 45 hr. Exchange was quenched with 1:1 (vol:vol quench buffer: deuterated-protein buffer solution) cold quench buffer (100 mM phosphate pH 2.4, 0.5 M TCEP, 3 M guanidium chloride) to pH 2.4. The quenched sample was passed through a home-made immobilized pepsin column for digestion. The resulting peptides were trapped and desalted on a small C8 column (Higgins Analytical TARGA C8 5 µm 5× 1.0 mm). After desalting (3 min at 0°C) peptides were eluted by a gradient (8 µL/min, 10% to 40% acetonitrile over 15 min) and passed through an analytical column (Higgens Analytical TARGA C8 5 µm, 50× 0.3 mm) and introduced into a THERMO Q-Exactive mass spectrometer by electrospray (*Mayne et al., 2011*; *Mayne, 2016*).

Peptides were identified by MS/MS analysis of nondeuterated samples. MS/MS data was analyzed by SEQUEST (Thermo Proteome Discoverer) using a sequence database including BRAF NT2, BRAF NT3, HRAS, KRAS, pepsin, and many potential contaminants and decoy proteins. 110 and 183 peptides (BRAF NT2 and NT3, respectively [not including MBP-tag or linker peptides]) were identified by MS/MS of which 35 and 80 (BRAF NT2 and NT3, respectively) were consistently found with good intensity in HX runs and are used here. Deuterated samples were analyzed using ExMS2 (*Kan et al., 2019*).

## OpenSPR

### BRAF:RAS interaction

Binding studies of BRAF to RAS were measured using OpenSPR (Nicoya). BRAF NT1–4 and FL-BRAF were immobilized to at least 2500 RUs on Ni-NTA sensor (Nicoya) following the manufacturer's protocols in buffer (20 mM HEPES pH 7.4, 150 mM NaCl, 0.05% Tween-20). After immobilization, buffer switched to 20 mM HEPES pH 7.4, 150 mM NaCl, 0.23% glycerol, 0.05% Tween-20 for analyte injections (1% BSA added for KRAS injections to reduce non-specific binding). H/KRAS was flowed over at 30 µL/min for 10 min at increasing concentrations and the chip was regenerated with 10 mM NaOH at 150 µL/min, allowing 3 min for baseline stabilization after each RAS injection. For slow flow NT1:HRAS experiments, HRAS was flowed over at 5 µL/min for 30 min. Binding kinetics were determined by 1:1 fitting model and experiments plotted against each other using TraceDrawer software. $K_D$ values are reported as mean ± standard deviation.

### BRAF-KD:NT interaction

Binding studies of BRAF-KD-WT/D459G to BRAF NTs were measured using OpenSPR (Nicoya). BRAF-KD-WT/D594G was immobilized (50 µg/mL; ~6000 RUs) on a carboxyl sensor chip (Nicoya) following

the standard manufacturer's protocols of the amine coupling kit (Nicoya). Analyte (BRAF NT at 5, 15, 44, 133, 400 nM) was flowed over the sensor chip in buffer (20 mM HEPES pH 7.4, 150 mM NaCl, 5% glycerol, 0.005% Tween-20, 1% BSA) at a flow rate of 30 µL/min, allowing time for dissociation (10 min), to obtain real-time binding data. To disrupt the interaction, 400 nM BRAF NT1 was pre-bound with 400 nM HRAS, by rotating at 4°C for 1 hr, subsequently 3× diluted in HEPES buffer and flowed over the sensor in the same flow conditions. Binding kinetics were determined by 1:1 fitting model and experiments plotted against each other using TraceDrawer software. $K_D$ values are reported as mean ± standard deviation.

## Acknowledgements

This work was supported by WW Smith Charitable Fund (ZW), NIH R15GM128099 (ZW), and NIH R01GM138671 (ZW). Thanks to Dr. Leland Mayne at the University of Pennsylvania for technical support and data analysis of HDX-MS.

## Additional information

### Funding

| Funder | Grant reference number | Author |
|---|---|---|
| WW Smith Charitable Trust | | Zhihong Wang |
| National Institute of General Medical Sciences | R15GM128099 | Zhihong Wang |
| National Institute of General Medical Sciences | R01GM138671 | Zhihong Wang |

The funders had no role in study design, data collection and interpretation, or the decision to submit the work for publication.

### Author contributions

Tarah Elizabeth Trebino, Data curation, Formal analysis, Validation, Methodology, Writing - original draft, Writing - review and editing; Borna Markusic, Haihan Nan, Shrhea Banerjee, Data curation; Zhihong Wang, Conceptualization, Resources, Supervision, Funding acquisition, Methodology, Writing - original draft, Writing - review and editing

### Author ORCIDs

Tarah Elizabeth Trebino ⓘ http://orcid.org/0009-0008-6320-295X
Borna Markusic ⓘ http://orcid.org/0009-0007-8302-812X
Shrhea Banerjee ⓘ http://orcid.org/0009-0001-2929-4490
Zhihong Wang ⓘ http://orcid.org/0000-0003-1667-3536

Reviewer #1 (Public Review): https://doi.org/10.7554/eLife.88836.3.sa1
Reviewer #2 (Public Review): https://doi.org/10.7554/eLife.88836.3.sa2
Author Response https://doi.org/10.7554/eLife.88836.3.sa3

## Additional files

### Supplementary files
• MDAR checklist

• Supplementary file 1. Complete set of peptide plots. Time-dependent deuterium uptake plots for NT2+/- HRAS, NT2 +/-KRAS, and NT3+/-HRAS.

### Data availability
All data generated or analyzed during this study have been included in the manuscript and supporting files. Source data files have been provided for Figures 1, 2, 3, 4, 5, and 6.

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

# Appendix 1

**Appendix 1—key resources table**

| Reagent type (species) or resource | Designation | Source or reference | Identifiers | Additional information |
|---|---|---|---|---|
| Strain, strain background (*Escherichia coli*) | BL21 codon + | NEB | | |
| Strain, strain background (*Escherichia coli*) | DH5α | NEB | C2987I | |
| Antibody | Anti-GST (mouse monoclonal) | Santa Cruz Biotechnology | SC-138 | WB (1:5000) |
| Antibody | Anti-RAS (rabbit monoclonal) | Cell Signaling | 67648S | WB (1:5000) |
| Antibody | Anti-His (rabbit monoclonal) | Sigma-Aldrich | SAB5600227 | WB (1:5000) |
| Recombinant DNA reagent | GST-HRAS | Addgene | 55653 | |
| Recombinant DNA reagent | His/MBP-KRAS | Addgene | 159546 | |
| Sequence-based reagent | GST-KRAS_For | This paper | PCR primers | 5'ATCTGGTTCCGCGTGGA TCCACTGAATATAAACTTGTGGTAG 3' |
| Sequence-based reagent | GST-KRAS_Rev | This paper | PCR primers | 5' CAGTCAGTCACG ATGAATTCTTACATAATTACACACTTTGTCTTTG 3' |
| Sequence-based reagent | GST-vector_For | This paper | PCR primers | 5' GAATTCATCGTGACTGACTGACG 3' |
| Sequence-based reagent | GST-vector_Rev | This paper | PCR primers | 5' GGATCCACGCGGAACCAG 3' |
| Sequence-based reagent | BRAF-RBD_For 1 | This paper | PCR primers | 5' CATATGCTCGGATCCGCGG CGCTGAGCGGTG 3' |
| Sequence-based reagent | BRAF-RBD_For 151 | This paper | PCR primers | 5' CATATGCTCGGATCCTCACCA CAAAAACCTATCGTTAG 3' |
| Sequence-based reagent | BRAF-RBD_Rev 227 | This paper | PCR primers | 5' GTTGTAAGAATTCAAGCTTAC AACACTTCCACATGCAATTC 3' |
| Sequence-based reagent | BRAF-RBD_Rev 288 | This paper | PCR primers | 5' CAAAGAACTGAATT CAAGCTTACAAATCAAGTTGGT 3' |
| Sequence-based reagent | BRAF-CRD_For | This paper | PCR primers | 5' AGCAAATGGGTCGCGGATCCA CACACAACTTTGTACGAAAAAC'3 |
| Sequence-based reagent | BRAF-CRD_Rev | This paper | PCR primers | 5'CGAGTGCGGCCGCAAGCTTAC AAATCAAGTTGGTCATAATTAAC'3 |
| Commercial assay or kit | EZ-Link Sulfo-NHS-Biotin, No-Weigh Format | Thermo Fisher | A39256 | |
| Chemical compound, drug | GMPPNP | Sigma-Aldrich | G0635 | |
| Software, algorithm | ExMS2 | https://doi.org/10.1021/acs.analchem.9b01682 | | |
| Software, algorithm | TraceDrawer | https://tracedrawer.com/ | | |
| Software, algorithm | SEQUEST | Thermo Proteome Discoverer | | |

