## [Editor Report · eLife assessment]

This manuscript describes **useful** information on the interactions of the BRAF N-terminal regulatory regions (CRD, RBD and BSR) with the C-terminal kinase domain and with the upstream regulators HRAS and KRAS. The authors provide **solid** evidence that the BRAF BSR domain may negatively regulate RAS binding and propose that the presence of the BSR domain in BRAF provides an additional layer of autoinhibitory constraints. The data will be of interest for researchers in the RAS/RAF and general kinase regulation fields.

---

## [Referee Report · Reviewer #1 (Public Review)]

Trebino et al. investigated the BRAF activation process by analysing the interactions of BRAF N-terminal regulatory regions (CRD, RBD and BSR) with the C-terminal kinase domain and with the upstream regulators HRAS and KRAS. To this end, they generated four constructs comprising different combinations of N-terminal domains of BRAF and analysed their interaction with HRAS as well as conformational changes that occur. By HDX-MS they confirmed that the RBD is indeed the main mediator of interaction with HRAS. Moreover, they observed that HRAS binding leads to conformational changes exposing the BSR to the environment. Next, the authors used OpenSPR to determine the binding affinities of HRAS to the different BRAF constructs. While BSR+RBD, RBD+CRD and RBD bound HRAS with nanomolar affinity, no binding was observed with the construct comprising all three domains. Based on these experiments, the authors concluded that BSR and CRD negatively regulate binding to HRAS and hypothesised that BSR may confer some RAS isoform specificity. They corroborated this notion by showing that KRAS bound to BRAF-NT1 (BSR+RBD+CRD) while HRAS did not. Next, the authors analysed the autoinhibitory interaction occurring between the N-terminal regions and the kinase domain. Through pulldown and OpenSPR experiments, they confirm that it is mainly the CRD that makes the necessary contacts with the kinase domain. In addition, they show that the BSR stabilizes these interactions and that the addition of HRAS abolishes them. Finally, the D594G mutation within the KD of BRAF is shown to destabilise these autoinhibitory interactions, which could explain its oncogenic potential.

Overall, the in vitro study provides new insights into the regulation of BRAF and its interactions with HRAS and KRAS through a comprehensive in vitro analysis of the BRAF N-terminal region. Also, the authors report the first KD values for the N- and C-terminal interactions of BRAF and show that the BSR might provide isoform specificity towards KRAS. While these findings could be useful for the development of a new generation of inhibitors, the overall impact of the manuscript could probably be enhanced if the authors were to investigate in more detail how the BSR-mediated specificity of BRAF towards certain RAS isoforms is achieved. Moreover, though the very "clean" in vitro approach is appreciated, it also seems useful to examine whether the observed interactions and conformational changes occur in the full-length BRAF molecule and in more physiological contexts. Some of the results could be compared with studies including full length constructs.

---

## [Referee Report · Reviewer #2 (Public Review)]

In the manuscript the authors conduct a series of in vitro experiments using N-terminal and C-terminal BRAF fragments (SPR, HDX-MS, pull-down assays) to interrogate BRAF domain-specific autoinhibitory interactions and engagement by H- and KRAS GTPases. Of the three RAF isoforms, BRAF contains an extended N-terminal domain that has yet to be detected in X-ray and cryoEM reconstructions but has been proposed to interact with the KRAS hypervariable region. The investigators probe binding interactions between 4 N-terminal (NT) BRAF fragments (containing one more NT domain (BRS, RBD, and CRD)), with full-length bacterial expressed HRAS, KRAS as well as two BRAF C-terminal kinase fragments to tease out the underlying contribution of domain-specific binding events. They find, consistent with previous studies, that the BRAF BSR domain may negatively regulate RAS binding and propose that the presence of the BSR domain in BRAF provides an additional layer of autoinhibitory constraints that mediate BRAF activity in a RAS-isoform-specific manner. One of the fragments studied contains an oncogenic mutation in the kinase domain (BRAF-KDD594G). The investigators find that this mutant shows reduced interactions with an N-terminal regulatory fragment and postulate that this oncogenic BRAF mutant may promote BRAF activation by weakening autoinhibitory interactions between the N- and C-terminus.

The manuscript is now significantly improved. The inclusion of additional controls and new experiments with KRAS strengthen the manuscript and aid in establishing RAS isoform-specific BRAF interactions.

---

## [Author Response]

The following is the authors’ response to the original reviews.

**Public Reviews**:
**Reviewer #1 (Public Review):**
Trebino et al. investigated the BRAF activation process by analysing the interactions of BRAF N-terminal regulatory regions (CRD, RBD, and BSR) with the C-terminal kinase domain and with the upstream regulators HRAS and KRAS. To this end, they generated four constructs comprising different combinations of N-terminal domains of BRAF and analysed their interaction with HRAS as well as conformational changes that occur. By HDX-MS they confirmed that the RBD is indeed the main mediator of interaction with HRAS. Moreover, they observed that HRAS binding leads to conformational changes exposing the BSR to the environment. Next, the authors used OpenSPR to determine the binding affinities of HRAS to the different BRAF constructs. While BSR+RBD, RBD+CRD, and RBD bound HRAS with nanomolar affinity, no binding was observed with the construct comprising all three domains. Based on these experiments, the authors concluded that BSR and CRD negatively regulate binding to HRAS and hypothesised that BSR may confer some RAS isoform specificity. They corroborated this notion by showing that KRAS bound to BRAF-NT1 (BSR+RBD+CRD) while HRAS did not. Next, the authors analysed the autoinhibitory interaction occurring between the N-terminal regions and the kinase domain. Through pulldown and OpenSPR experiments, they confirm that it is mainly the CRD that makes the necessary contacts with the kinase domain. In addition, they show that the BSR stabilizes these interactions and that the addition of HRAS abolishes them. Finally, the D594G mutation within the KD of BRAF is shown to destabilise these autoinhibitory interactions, which could explain its oncogenic potential.Overall, the in vitro study provides new insights into the regulation of BRAF and its interactions with HRAS and KRAS through a comprehensive in vitro analysis of the BRAF N-terminal region. Also, the authors report the first KD values for the N- and C-terminal interactions of BRAF and show that the BSR might provide isoform specificity towards KRAS. While these findings could be useful for the development of a new generation of inhibitors, the overall impact of the manuscript could probably be enhanced if the authors were to investigate in more detail how the BSR-mediated specificity of BRAF towards certain RAS isoforms is achieved. Moreover, though the very "clean" in vitro approach is appreciated, it also seems useful to examine whether the observed interactions and conformational changes occur in the full-length BRAF molecule and in more physiological contexts. Some of the results could be compared with studies including full-length constructs.

Public Response: We would like to express our gratitude for your valuable feedback on our manuscript. Your insightful suggestions have significantly improved the quality and completeness of our research. In response to your comments, we have conducted additional experiments and incorporated new data into the revised manuscript.

To gain a deeper understanding of how the BSR-mediated specificity of BRAF towards certain RAS isoforms is achieved, we performed HDX-MS to investigate the impact of KRAS interactions on the BSR. Our findings indicate that when KRAS is bound to BRAF NT2, there is no significant difference in hydrogen-deuterium exchange rates in the BSR compared to the apo-NT2 state (Figure 4). This observation contrasts with the effect of HRAS binding, where peptides from the BRAF-BSR exhibit an increased rate change, suggesting that HRAS induces a conformationally more dynamic state (Figure 2).

Our results align with the conclusions of Terrell et al. in their 2019 publication, which propose that isoform preferences in the RAS-RAF interaction are driven by opposite charge attractions between BRAF-BSR and KRAS-HVR, promoting the interaction.1 Our data offers a potential mechanistic explanation, suggesting that HRAS disrupts the conformational stability of the BSR provided by the RBD, while KRAS-HVR restores stability and enhances interaction favorability. It is important to note that our results do not directly confirm a long-lasting interaction between the BRAF-BSR and KRAS-HVR, but they do not rule out the possibility of a transient, low-affinity interaction or close proximity between the two.

Furthermore, our binding kinetics measurements conducted using OpenSPR support these findings. Particularly, in the case of NT1, when the CRD accompanies the BSR and RBD, no interactions with HRAS were observed. Additionally, we quantified the binding affinities between NT3:KRAS and NT4:KRAS, demonstrating that they are equally strong and that the presence of the BSR or CRD does not singularly affect the primary RBD interaction, consistent with HRAS. The BSR appears to exert an inhibitory effect on HRAS when the entire N-terminal region (BSR+RBD+CRD) is present. The BSR-mediated specificity is achieved through a coordinated interplay with the CRD.

Moreover, we have addressed your concern regarding the physiological relevance of our conclusions. In response, we utilized active, full-length (FL) BRAF purified from HEK293F cells in OpenSPR experiments. Our findings indicate that FL-BRAF behaves similarly to BRAF-NT1, as it does not bind to HRAS but binds to KRAS with a deviation comparable to NT1. We have demonstrated that post-translational modifications or native intramolecular interactions do not alter our initial results. Several literature sources, employing cell systems or expressing proteins from insect or mammalian cells, further support the findings presented in our study.2–5

Thank you once again for your constructive feedback, which has contributed significantly to the refinement of our work.

For the author:Major points:1. Figure 1D: Negative control is missing.

Response: We have incorporated the negative control into this figure as suggested.

2. Figure 3F and G: negative controls (GST only) are missing.

Response: We have incorporated the negative control into this figure as suggested.

3. The authors demonstrate that BRAF NT1 (BSR+RBD+CRD) interacts with KRAS but not HRAS in SPR experiments (Figure 4). What about the conformational change that affects the positioning of BSR when NT2 (BSR+RBD) binds to HRAS (Figure 2)? Does it also occur with KRAS or not? When a rate change is observed between free protein and bound protein in HDX, particularly when this rate change results in a sigmoidal curve that closely parallels the reference curve, it signifies that all residues within the peptide share a uniform protection factor. This suggests that they collectively undergo conformational changes at the same rate, likely due to a concerted opening as a cohesive unit. In the context of our time plots, we observe this distinctive characteristic in the curves derived from the BSR peptides, indicating that HRAS binding perturbs this region, alters its flexibility, and induces a coordinated conformational shift. This compelling evidence strongly supports our assertion that HRAS instigates a reorientation of the BSR.

Response: In response to the reviewer's comments, we conducted additional experiments to explore whether KRAS elicits any comparable alterations in the H-D exchange of the BSR within BRAF-NT2. Our findings indicate that KRAS does not induce a similar conformational change in the BSR. We have detailed these results in the Results section under the heading "BSR Differentiates the BRAF-KRAS Interaction from the BRAF-HRAS Interaction" and have included corresponding panels in Figure 4 to visually illustrate these observations.

4. Related to point 3: The authors mention that the HVR domain is responsible for isoform-specific differences. Does the BSR interact with the HVR domain of KRAS (but not HRAS)?

Response: It has been suggested by Terrell and colleagues1 that the BRAF-BSR and KRASHVR are directly responsible for the isoform specific interactions. We have no direct evidence confirming an interaction between the HVR and BSR. However, we deduce the possibility of such interaction based on previous research findings. Our HDX-MS experiments have demonstrated that the BRAF-BSR does not engage with HRAS. In our new HDX-MS experiments involving KRAS, we observed that the presence of KRAS does not lead to any discernible increase or decrease in the rate of deuterium exchange within the BRAF-BSR. It is important to emphasize that the absence of a rate change does not necessarily negate the occurrence of binding; rather, it might indicate a transient interaction with an affinity level below the detection threshold of HDX-MS.

Given that the only major difference between H- and K-RAS isoforms is the HVR, we hypothesize that binding differences between BRAF and RAS isoforms can be attributed to the HVR. Notably, BRAF-NT3 resembles CRAF, which also behaves in line with the findings from Terrell et al. in which the BSR is not present to impact RAS-RAF association. We have updated some of the discussion section to include the new results and draw relevant conclusion.

We mention in the text in the results section, “The HVR is an important region for regulating RAS isoform differences, like membrane anchoring, localization, RAS dimerization, and RAF interactions6… These results, combined with HDX-MS results, which showed that theBSR is exposed when bound to HRAS, suggest that the electrostatic forces surrounding theBSR promote BRAF autoinhibition and the specificity of RAF-RAS interactions.”

We also write in the discussion, “However, BRET assays suggest that CRAF does not show preference for either H- or KRAS, while BRAF appears to prefer KRAS.1 This preference is suggested to result from the potential favorable interactions between the negatively charged BSR of BRAF and the positively charged, poly-lysine region of the HVR of KRAS1… Our binding data provide additional examples of isoform-specific activity. We speculate that diminished BRAF-NT1 binding to HRAS and increased BSR exposure upon HRAS binding may be due to electrostatic repulsion between HRAS and the BSR. Our full-length KRAS and its interaction with NT1 support the hypothesis that the BSR attenuates fast binding to HRAS but not to KRAS.”

5. The authors might consider including NRAS in their study to give more weight to this interesting aspect.

Response: While this suggestion is intriguing and could contribute to the expanding body of literature on RAS signaling, particularly in the context of NRAS-mutant tumors, we believe that delving into this topic would be beyond the scope of the present manuscript.

6. Figure 6A: In this pulldown experiment the authors wish to demonstrate that binding of HRAS abolishes the autoinhibitory binding between NT1 and the kinase domain. However, the experimental design (i.e., pulldown of RAS) does not allow us to assess whether NT1 and KD are bound to each other in these conditions at all. The authors should rather pull down the KD and show that the interaction with NT1 is abolished when RAS is added.

Response: We appreciate your suggestion. The experimental design for this study was intentionally structured to focus on the specific subset of NT1 that interacts with HRAS. The BRAF N-terminal region has the capacity to bind both HRAS and KD, resulting in two distinct populations within BRAF-NT1: NT1:KD and NT1:HRAS, although we believe the ratio between those two populations is not 1:1. If we were to design the experiment by isolating either the KD or NT1, it would lead to the observation of both populations simultaneously, making it challenging to distinguish between them. Our pulldown experiments are performed under the same conditions (i.e. all the proteins were maintained in a molar ratio of 1:1 and exposed to the same buffer components), and we rely on pulldown assays, such as those depicted in Figure 5, to clearly demonstrate the binding interactions between NT1 and KD.

7. The authors have chosen a purely in vitro approach for their interaction studies, which initially makes sense for the addressed questions. However, since the BRAF constructs studied are only fragments and neither BRAF nor K/HRAS has any posttranslational modifications, the question arises to what extent the findings obtained hold up in vivo. Therefore, the manuscript would greatly benefit from monitoring the described interactions in full-length proteins and in cells or at least with proteins purified from cells.

Response: Thank you for your valuable suggestion, which we take very seriously to enhance the quality of our manuscript. Upon carefully reviewing your comments, we conducted additional experiments involving full-length, wild-type BRAF (FL-BRAF) that was purified from mammalian cells, encompassing the post-translational modifications and scaffolding proteins such as 14-3-3 (Supplementary Fig 8A). We have incorporated the findings from these OpenSPR experiments into the revised manuscript within the Results Section titled "BSR Differentiates the BRAF-KRAS Interaction from the BRAFHRAS Interaction" and Figure 4. In summary, our results with FL-BRAF affirm the extension of our initial observations. Both NT1 and FL-BRAF interact with KRAS with comparable affinities, and neither NT1 nor FL-BRAF demonstrates an interaction with HRAS using OpenSPR. These results underscore that BRAF fragments accurately represent active, fully processed BRAF, lending support to our in vitro approach.

Moreover, the conserved interactions we report in this manuscript are supported by literature. The interaction between RAF-RBD and RAS has been extensively documented, spanning investigations conducted in both insect and mammalian cell lines. For instance, Tran et al. (2021) utilized mammalian expression systems to explore the role of RBD in mediating BRAF activation through RAS interaction, identifying the same binding surfaces that we highlighted using HDX-MS.2 They quantified the KRAS-CRAF interaction yielding binding affinities in the low nanomolar range, similar to our findings for BRAF-NT:KRAS OpenSPR.2 In the manuscript text, we compared the binding affinity of BRAF residues 1245 purified from insect cells3 to our BRAF 1-227 (NT2 from *E. coli*), noting that the published value falls within the standard deviation of our experimental value. Additionally, our results align with the autoinhibited FL-BRAF:MEK:14-3-3 structure, which was expressed in Sf9 insect cells and reveals the central role of the CRD in maintaining autoinhibition through interactions with KD.4 In 2005, Tran and colleagues revealed specific domains within the BRAF N-terminal region are involved in binding to KD through Co-IP experiments conducted in mammalian cells.5

While we are fully aware of the limitations of taking a purely in vitro approach to study the role of BRAF regulatory domains in RAS-RAF interactions and autoinhibition, as well as to quantify the affinity of these interactions, we emphasize that this approach enables us to dissect and examine the specific regions of RAF that are under investigation. As we write in the manuscript: “Our in vitro studies were conducted using proteins purified from *E. coli*, which lack the membrane, post-translational modifications, and regulatory, scaffolding, or chaperone proteins that are involved in BRAF regulation. Nonetheless, our study provides a direct characterization of the intra- and inter-molecular protein-protein interactions involved in BRAF regulation, without the complications that arise in cell-based assays.” We have added the following comment to clarify the advantages of our in vitro approach and the challenges associated with cell-based assays: “… without the complications and false-positives that can arise in cell-based assays, which often cannot distinguish between proximity and biochemical interactions.”

Once again, we appreciate your insight feedback, which has contributed significantly to the improvement of our manuscript.

Minor:1. Page 7, paragraph 2, line 6: It should probably read "BRAF autoinhibition" not "BRAF autoinhibitory".

Response: Thank you for bringing this to our attention. We have fixed this typo.

2. Figure 3G: In the first lane (time point 0 min) there is no input band for His/MBP-NT1. Probably a mistake when cropping the image from the original photo.

Response: We sincerely appreciate your diligence in identifying cropping errors, and we have taken comprehensive measures to review the manuscript and correct any such errors. Regarding this specific figure, it is important to note that NT1 was not added at the "0" minute time point, which explains the absence of an input band at that stage. To avoid any confusion, we have revised the notation from "0" to "-" for clarity.

**Reviewer #2 (Public Review):**
In the manuscript entitled 'Unveiling the Domain-Specific and RAS Isoform-Specific Details of BRAF Regulation', the authors conduct a series of in vitro experiments using Nterminal and C-terminal BRAF fragments (SPR, HDX-MS, pull-down assays) to interrogate BRAF domain-specific autoinhibitory interactions and engagement by H- and KRAS GTPases. Of the three RAF isoforms, BRAF contains an extended N-terminal domain that has yet to be detected in X-ray and cryoEM reconstructions but has been proposed to interact with the KRAS hypervariable region. The investigators probe binding interactions between 4 N-terminal (NT) BRAF fragments (containing one more NT domain (BRS, RBD, and CRD)), with full-length bacterial expressed HRAS, KRAS as well as two BRAF C-terminal kinase fragments to tease out the underlying contribution of domainspecific binding events. They find, consistent with previous studies, that the BRAF BSR domain may negatively regulate RAS binding and propose that the presence of the BSR domain in BRAF provides an additional layer of autoinhibitory constraints that mediate BRAF activity in a RAS-isoform-specific manner. One of the fragments studied contains an oncogenic mutation in the kinase domain (BRAF-KDD594G). The investigators find that this mutant shows reduced interactions with an N-terminal regulatory fragment and postulate that this oncogenic BRAF mutant may promote BRAF activation by weakening autoinhibitory interactions between the N- and C-terminus.While this manuscript sheds light on B-RAF specific autoinhibitory interactions and the identification and partial characterization of an oncogenic kinase domain (KD) mutant, several concerns exist with the vitro binding studies as they are performed using taggedisolated bacterial expressed fragments, 'dimerized' RAS constructs, lack of relevant citations, controls, comparisons and data/error analysis. Detailed concerns are listed below.1. Bacterial-expressed truncated BRAF constructs are used to dissect the role of individual domains in BRAF autoinhibition. Concerns exist regarding the possibility that bacterial expression of isolated domains or regions of BRAF could miss important posttranslational modifications, intra-molecular interactions, or conformational changes that may occur in the context of the full-length protein in mammalian cells. This concern is not addressed in the manuscript.

Response: Reviewer 1 raised a similar concern, and we have duplicated our response below for your reference:

Thank you for your valuable suggestion, which we take very seriously to enhance the quality of our manuscript. Upon carefully reviewing your comments, we conducted additional experiments involving full-length, wild-type BRAF (FL-BRAF) that was purified from mammalian cells, encompassing the post-translational modifications and scaffolding proteins such as 14-3-3 (Supplementary Fig 8A). We have incorporated the findings from these OpenSPR experiments into the revised manuscript within the Results Section titled "BSR Differentiates the BRAF-KRAS Interaction from the BRAF-HRAS Interaction" and Figure 4. In summary, our results with FL-BRAF affirm the extension of our initial observations. Both NT1 and FL-BRAF interact with KRAS with comparable affinities, and neither NT1 nor FL-BRAF demonstrates an interaction with HRAS using OpenSPR. These results underscore that BRAF fragments accurately represent active, fully processed BRAF, lending support to our in vitro approach.

Moreover, the conserved interactions we report in this manuscript are supported by literature. The interaction between RAF-RBD and RAS has been extensively documented, spanning investigations conducted in both insect and mammalian cell lines. For instance, Tran et al. (2021) utilized mammalian expression systems to explore the role of RBD in mediating BRAF activation through RAS interaction, identifying the same binding surfaces that we highlighted using HDX-MS.2 They quantified the KRAS-CRAF interaction yielding binding affinities in the low nanomolar range, similar to our findings for BRAF-NT:KRAS OpenSPR.2 In the manuscript text, we compared the binding affinity of BRAF residues 1245 purified from insect cells3 to our BRAF 1-227 (NT2 from *E. coli*), noting that the published value falls within the standard deviation of our experimental value. Additionally, our results align with the autoinhibited FL-BRAF:MEK:14-3-3 structure, which was expressed in Sf9 insect cells and reveals the central role of the CRD in maintaining autoinhibition through interactions with KD.4 In 2005, Tran and colleagues revealed specific domains within the BRAF N-terminal region are involved in binding to KD through Co-IP experiments conducted in mammalian cells.5

While we are fully aware of the limitations of taking a purely in vitro approach to study the role of BRAF regulatory domains in RAS-RAF interactions and autoinhibition, as well as to quantify the affinity of these interactions, we emphasize that this approach enables us to dissect and examine the specific regions of RAF that are under investigation. As we write in the manuscript: “Our in vitro studies were conducted using proteins purified from *E. coli*, which lack the membrane, post-translational modifications, and regulatory, scaffolding, or chaperone proteins that are involved in BRAF regulation. Nonetheless, our study provides a direct characterization of the intra- and inter-molecular protein-protein interactions involved in BRAF regulation, without the complications that arise in cell-based assays.” We have added the following comment to clarify the advantages of our in vitro approach and the challenges associated with cell-based assays: “… without the complications and false-positives that can arise in cell-based assays, which often cannot distinguish between proximity and biochemical interactions.”

Once again, we appreciate your insight feedback, which has contributed significantly to the improvement of our manuscript.

2. The experiments employ BRAF NT constructs that retain an MBP tag and RAS proteins with a GST tag. Have the investigators conducted control experiments to verify that the tags do not induce or perturb native interactions?

Response: Thank you for highlighting this important issue. We have conducted control experiments whenever feasible, particularly in cases where tags were not required for visualization, immobilization, or where cleave sites were present. We have subsequently included these control experiments in the supplementary figures and accompanying text within the manuscript.

It is essential to note that many of the techniques employed in this manuscript rely on tags, such as immobilizing proteins onto NTA OpenSPR sensors and employing various resins/beads for pulldown assays. Utilizing tags for protein immobilization in OpenSPR applications offers distinct advantages, including homogeneous and site-specific immobilization of the protein, ensuring that binding sites remain accessible for the study of protein-protein interactions (PPIs) of interest. Furthermore, in all BRAF-RAS SPR experiments, the MBP protein serves as the reference channel "blocking" protein. This reference channel is instrumental in mitigating any potential false-positive signals resulting from binding interactions with the MBP protein. Any such signal is subsequently subtracted out during data analysis.

To provide a comprehensive understanding of these aspects, we have incorporated these details into the manuscript text for clarity:

“Maltose bind protein (MBP) is immobilized on the OpenSPR reference channel, which accounts for any non-specific binding or impacts to the native PPIs that may result from the presence of tags. Kinetic analysis is performed on the corrected binding curves, which subtracts any response in the reference channel.”

We describe the control experiment to examine whether His/MBP-tag affects NT1 binding with BRAF-KD: “Similarly, we removed the His/MBP-tag from BRAF-NT1 through a TEV protease cleavage reaction and flowed over untagged NT1. Kinetic analysis confirmed that the interaction is preserved with the KD=13 nM (Supplemental Figure 6F).”

We show that the GST-tag does not affect KRAS interactions with NTs in supplemental figure 6. We purified full-length, His/MBP-KRAS and subsequently removed the tag through TEV cleavage. BRAF-NT interactions are preserved with untagged KRAS. GST alone, also does not interact with BRAF-NTs. We updated the text in the results section “BSR differentiates the BRAF-KRAS interaction from the BRAF-HRAS interaction.”

Additionally, Vojtek and colleagues used the same fusion-protein combinations (GSTRAS and MBP-RAF) in pulldown experiments and also found no perturbations from these tags.8

3. The investigators state that the GST tag on the RAS constructs was used to promote RAS dimerization, as RAS dimerization is proposed to be key for RAF activation. However, recent findings argue against the role of RAS dimers in RAF dimerization and activation (Simanshu et al, Mol. Cell 2023). Moreover, while GST can dimerize, it is unclear whether this promotes RAS dimerization as suggested. In methods for the OpenSPR experiments probing NT BRAF:RAS interactions, it is stated that "monomeric KRAS was flowed...". This terminology is a bit confusing. How was the monomeric state of KRAS determined and what was the rationale behind the experiment? Is there a difference in binding interactions between "monomeric vs dimeric KRAS"?

Response: Thank you for conducting such a comprehensive review of our manuscript and for identifying the mention of "monomeric KRAS" in the experimental section, which was inadvertently included and should not have been present. This terminology originally referred to a series of experiments involving "monomeric" KRAS that were initially considered for inclusion in the main body of the manuscript but were subsequently removed before submission. Furthermore, we adjusted the terminology to prevent any confusion or unwarranted implications.

To clarify, this "monomeric" construct refers to the tagless, full-length KRAS variant that was confirmed to exist in a monomeric state through Size Exclusion Chromatography, eluting at a volume equivalent to 21 kDa. We have incorporated the findings from experiments involving this untagged KRAS variant into the supplementary figures to provide supporting evidence, particularly in response to comment #2, that the GST-tag does not interfere with native interactions. Supplementary Figure 1 illustrates that both GST-HRAS (45 kDa) and GST-KRAS (45 kDa) elute as dimers in solution, at approximately 90 kDa. It is important to note that the main text figures primarily feature the GST-tagged, "dimeric" RAS constructs. Our research results do not suggest any significant differences between "monomeric," untagged KRAS and "dimeric" GST-tagged KRAS, indicating that the binding kinetics between RAS and RAF are not influenced by oligomerization state (Supplementary Fig 6). To mitigate any potential confusion, we have made the necessary distinctions in the text and have revised the methods description to accurately reflect these aspects.

While the recent findings summarized by Simanshu and colleagues were published concurrently with our manuscript submission, we would like to address this comment in the following manner. The authors assert that RAS does not engage in dimerization through the G domain, a hypothesis that contrasts with certain prior research findings. Instead, they propose that the plasma membrane plays a pivotal role in the clustering of RAS. Furthermore, the authors mention the involvement of RAS "dimerization" in RAF dimerization and activation in the subsequent statements:

“Recruitment of two RAF proteins by RAS proteins in close proximity facilitate RAF activation but are not required for RAF dimerization.”

“However, the PM recruitment of two RAF proteins by two non-dimerized but co- localized RAS proteins would serve equally well to promote RAF dimerization. Moreover, recent work on the activation cycle of RAF dimers (ref 20–23) argues strongly against a role for RAS dimers while revealing regulation by the 14-3-3 and SHOC2-MRAS- PP1C complexes. (Ref 24)”

The primary focus of our study centers on elucidating the intricate details of the RAS-RAF interaction and the mechanisms underlying RAF autoinhibition, rather than emphasizing RAF dimerization as the sole pathway to RAF activation. It is important to recognize that RAF activation encompasses multiple steps, including RAS-mediated relief of RAF autoinhibition.

To mimic physiological conditions as closely as possible, we employed a GST-tag on RAS in our experiments. It's worth noting that GST has a dimerization property,9 which brings RAS molecules into close proximity to one another, effectively emulating conditions akin to the plasma membrane. Our primary objective is not solely to facilitate interactions by bringing RAS into close proximity. Instead, our aim is to replicate cellular conditions to the greatest extent feasible, especially within the predominantly in vitro framework of our studies. Furthermore, we have revised the sentence pertaining to HRAS as follows: “As verified by size exclusion chromatography (Supplementary Fig 1A), the GST-tag dimerizes and forces HRAS into close proximity to recapitulate physiological conditions. (ref. 35)”

4. The investigators determine binding affinities between GST-HRAS and NT BRAF domains (NT2 7.5 {plus minus} 3.5; NT3 22 {plus minus} 11 nM) by SPR, and propose that the BRS domain has an inhibitory role HRAS interactions with the RAF NT. However, it is unclear whether these differences are statistically meaningful given the error.

Response: Thank you for bringing up this matter for further discussion. We are fully aware that these distinctions (NT2 and NT3), considering the overlapping error, lack statistical significance. Our conclusion points toward the most notable differences occurring when comparing NT1 to either NT2 or NT3, highlighting that the presence of the BSR has an inhibitory effect, particularly when the CRD is also present. It's important to note that we did not directly compare NT2 and NT3 to each other. Our comparison primarily elucidates that BSR without the CRD, and conversely, CRD without the BSR, do not exhibit the inhibitory effect. This collective evidence leads to the conclusion that all three domains collaboratively play a role in negatively regulating BRAF against HRAS.

5. It is unclear why NT1 (BSR+RBD+CRD) was not included in the HDX experiments, which makes it challenging to directly compare and determine specific contributions of each domain in the presence of HRAS. Including NT1 in the experimental design could provide a more comprehensive understanding of the interplay between the domains and their respective roles in the HRAS-BRAF interaction. Further, excluding certain domains from the constructs, such as the BSR or CRD, may overlook potential domain-domain interactions and their influence on the conformational changes induced by HRAS binding.

Response: We acknowledge that incorporating NT1 into the HDX experiments would have provided clearer insights into the specific contributions of each domain. Originally, it was our intention to include NT1 in these experiments. Unfortunately, we encountered challenges with the HDX experiments when it came to BRAF-NT1, as it yielded a significantly low sequence coverage after MS/MS analysis. We made multiple attempts to address this issue, which included additional protein purifications involving reducing agents, increasing the concentration of reaction buffer components, and extending the incubation time with reducing agents before injection. Despite these efforts, we were unable to obtain the desired sequence coverage for NT1. Consequently, we switched our approach to analyze NT2 and NT3 as the next best alternative.

6. The authors perform pulldown experiments with BRAF constructs (NT1: BSR+RBD+CRD, NT2: BSR+RBD, NT3: RBD+CRD, NT4: RBD alone), in which biotinylated BRAF-KD was captured on streptavidin beads and probed for bound His/MBP-tagged BRAF NTs. Western blot results suggest that only NT1 and NT3 bind to the KD (Figure 5). However, performing a pulldown experiment with an additional construct, CRD alone, it would help to determine whether the CRD alone is sufficient for the interaction or if the presence of the RBD is required for higher affinity binding. This additional experiment would strengthen the authors' arguments and provide further insights into the mechanism of BRAF autoinhibition.

Response: We are grateful for this valuable suggestion, and in response, we have taken the initiative to clone and purify a CRD-only construct (NT5) to strengthen our arguments. Subsequently, we conducted OpenSPR experiments to measure the binding affinity between NT5 and KD. Our findings clearly indicate that the CRD alone is not sufficient to mediate the autoinhibitory interactions and that the presence of the RBD is indeed necessary. These results have been incorporated into Figure 5 and are described within the Results Section for enhanced clarity and support.

7. While the investigators state that their findings indicate that H- and KRAS differentially interact with BRAF, most of the experiments are focused on HRAS, with only a subset on KRAS. As SPR & pull-down experiments are only conducted on NT1 and NT2, evidence for RAS isoform-specific interactions is weak. It is unclear why parallel experiments were not conducted with KRAS using BRAF NT3 & NT4 constructs.

Response: We sincerely appreciate your suggestion, which has contributed to enhancing the overall robustness of the evidence regarding isoform-specific differences between H- and K-RAS. In response, we performed additional experiments involving NT3 and NT4. The outcomes of these experiments have been integrated into Figure 4, and we have provided a comprehensive description of these results within the Results section “BSR differentiates the BRAF-KRAS interaction from the BRAF-HRAS interaction” of the manuscript.

8. The investigators do not cite the AlphaFold prediction of full-length BRAF (AFP15056-F1) or the known X-ray structure of the BRAF BRS domain. Hence, it is unclear how Alpha-Fold is used to gain new structural information, and whether it was used to predict the structure of the N-terminal regulatory or the full-length protein.

Response: We greatly appreciate the reviewer’s commitment to upholding good scientific practices and ensuring the inclusion of relevant citations in publications. In our original manuscript, we employed the UniProt ID P15056 to reference the specific AlphaFold structure used in our study. This was clarified as follows: "Since the full-length structure of BRAF is still unresolved, we applied the AlphaFold Protein Structure Database for a model of BRAF to display the conformation of the N-terminal domains and the HDX-MS results.40,41” Additionally, we referenced AlphaFold using the two citations recommended on their website (references 35 and 36 in the original manuscript). To prevent any potential confusion in the future, we have incorporated "AF-P15056-F1," as suggested.

We are sorry for any misunderstanding that may have arisen regarding the use of AlphaFold for gaining new structural insights. Our sole intention was to utilize AlphaFold as a tool for modeling HDX, as a full-length structure of BRAF, encompassing the entire N-terminal domain, remains unavailable. We have taken steps to clarify our objectives in the manuscript to ensure the purpose of our AlphaFold utilization is unambiguous.

Furthermore, we wish to emphasize that our utilization of AlphaFold was never intended to exclude the known X-ray structure of the BRAF-BSR domain. In our revised text, we have added clarity to our purposes and cited the Lavoie et al. Nature publication from 2018, which provides alignment between the X-ray structure and the AlphaFold model, thereby enhancing the confidence in the latter.

9. In HDX-MS experiments, it is unclear how the authors determine whether small differences in deuterium uptake observed for some of the peptide fragments are statistically significant, and why for some of the labeling reaction times the investigators state " {plus minus} HRAS only" for only 3 time points?

Response: First, in reference to the question about " ‘{plus minus} HRAS only’ for only 3 time points,” we write:

“Both constructs were incubated with and without GMPPNP-HRAS in D2O buffer for set labeling reaction times (NT3: 2 sec [NT3 ± HRAS only], 6 sec [NT3 ± HRAS only], 20 sec, 30 sec [NT3 ± HRAS only], 60 sec, 5 min, 10 min, 30 min, 90 min, 4.5 h, 15 h, and 24 h)...”

We realize how this can be confusing. To avoid such confusion, we fixed the text to read instead:

“Both constructs were incubated with and without GMPPNP-HRAS in D2O buffer for set labeling reaction times (NT3: 2 sec, 6 sec, 20 sec, 30 sec, 60 sec, 5 min, 10 min, 30 min, 90 min, 4.5 h, 15 h, 45 h and 24 h at RT; NT2: 20 sec, 60 sec, 5 min, 10 min, 30 min, 90 min, 4.5 h, 15 h, and 24 h at RT)...”

Next, with regard to assessing significance, we determine it by closely examining a consistent trend in smooth time course plots. To establish this trend, we rely on the presence of more than four overlapping peptides, each with multiple charge states, within a specific sequence range. When we observe multiple peptides showing even a small difference in rate exchange, we can confidently infer that structural changes have taken place. This confidence stems from the inherent reliability and redundancy in the data analysis approach we have employed.11,12 It is noteworthy that our focus is primarily on reporting the binding or no binding, rather than quantifying the magnitude of exchange. As such, conducting multiple replicates or statistical testing is not deemed necessary.13,14 This is true for multiple reasons:

1. Instead of small deuterium changes (y-axis), we are focusing on the x-axis changes, which provides a slowing factor and how much that H-D exchange rate has changed.

In a publication investigating the ideal HDX-MS data set, the author explains, “with the availability of high resolution HDX-MS raw data, it may be the time to shift the data analysis paradigm from determination of centroid values and presentation of deuteration levels to deconvolution of isotope envelopes and presentation of exchange rates.” 15Presentation of data through rate changes provides a physical chemistry measurement, as opposed to a relative measurement with percent deuteration. For example, slowing with a factor of 10 equates to the energy in 1 kCal. By quick visual estimation, we see a slowing factor of about 2 when RAS is bound to the BRAF-RBD.We made some changes to the text to clear up any confusion about measuring D uptake vs rate.

1. Looking at sigmoidal curves only—the “smooth time course” shows that the timedependent deuterium changes are not random, artifacts, or false positives/negatives. When parallel sigmoidal curves are present, any x-axis change is a measure of H-D exchange. Only plots with a smooth time course are used to make conclusions about BRAF’s conformational changes or binding interfaces.

2. Wide time range- the extended time also confirms that any observed difference is reliable and accurate. This extended time frame provides coverage for deuteration levels from 0 to 100% for peptides. A smooth time course is present in complete coverage.

A narrow time window is a common flaw in HDX-MS studies14,15

1. The rate change is observed at multiple time points (at least 4 for each peptide), which are all independent reactions, and show reproducibility of change

2. Many overlapping peptides show the same pattern- the exchange rate difference is observed in at least 4 peptide time plots without contradictory evidence within the sequence range.

We included the complete set of peptide time plots in the supplemental materials.

1. The many other peptide time plots that do not show any difference with and without RAS is a form of reproducibility, that no difference means no difference.

10. The investigators find that KRAS binds NT1 in SPR experiments, whereas HRAS does not. However, the pull-down assays show NT1 binding to both KRAS and HRAS. SI Fig 5 attributes this to slow association, yet both SPR (on/off rates) and equilibrium binding measurements are conducted. This data should be able to 'tease' out differences in association.

Response: Thank you for bringing up this important point. It's crucial to note that the experiments conducted at slow flow rates generated low responses, making it challenging to perform kinetic analyses effectively. Consequently, we are unable to provide accurate equilibrium binding measurements (on/off rates) for NT1 and HRAS. Regrettably, comparing the association rates between KRAS and HRAS is not feasible due to the differing flow rates employed. We have addressed this limitation in the manuscript as follows:

“We therefore immobilized NT1 and flowed over HRAS at a much slower flow rate(5 µL/min), during which we saw minimal but consistent binding (Supplementary Fig 5A). The low response and long timeframe of each injection, however, makes the dissociation constant (KD) unmeasurable and incomparable to our other NT-HRAS OpenSPR results.”

11. The model in Figure 7B highlights BSR interactions with KRAS, however, BSR interactions with the KRAS HVR (proximal to the membrane) are not shown, as supported by Terrell et al. (2019).

Response: Thank you for the suggestion. We reoriented the BSR closer to HVR of KRAS rather than G-domain.

12. The investigators state that 'These findings demonstrate that HRAS binding to BRAF directly relieves BRAF autoinhibition by disrupting the NT1-KD interaction, providing the first in vitro evidence of RAS-mediated relief of RAF autoinhibition, the central dogma of RAS-RAF regulation. However, in Tran et al (2005) JBC, they report pulldown experiments using N-and C-terminal fragments of BRAF and state that 'BRAF also contains an N-terminal autoinhibitory domain and that the interaction of this domain with the catalytic domain was inhibited by binding to active HRAS'. This reference is not cited.

Response: We appreciate the concern raised regarding our statement. We want to clarify that it was never our intention to disregard this JBC publication, and we apologize for any misunderstanding caused by our phrasing. We recognize that our initial statement was contentious, and we have removed the word "first" from the phrase "first in vitro evidence." In the section of the discussion where we originally cited the Tran et al. (2005) publication, we have revised the language to eliminate "first" and have rephrased the sentence, as provided below:

“Our in vitro binding studies align with previous implications that RAS relieves RAF autoinhibition shown through cell-based coIP’s.5”

13. In Fig 2, panels A and C, it is unclear what the grey dotted line in is each plot.

Response: Thank you for drawing our attention to the additional explanation needed here. The gray dotted lines represent the maximum deuterium exchange. We added the following description to the figure 2 legend:

“Gray dotted lines represent the theoretical exchange behavior for specified peptide that is fully unstructured (top) or for specified peptide with a uniform protection factor (fraction of time the residue is involved in protecting the H-bond) of 100 (lower).”

14. In Fig 3, error analysis is not provided for panel E.

Response: We added the standard deviation values to this panel. We additionally added these for Fig 4C and Fig 5B.

15. How was RAS GMPPNP loading verified?

Response: Ras loading is a well-established protocol with a solid foundation in the literature.16– 21 We followed this accepted method for nucleotide exchange. Our controls, as evident in pulldown and OpenSPR experiments (fig 1C, 4E), unequivocally demonstrate that GMPPNPloaded RAS is active, while unloaded RAS is inactive, as evidenced by the absence of no binding. We also added supplemental figure 6E to show that inactive (unloaded) GST-KRAS does not bind to BRAF during OpenSPR analysis. To exemplify this, we included binding curvesof 1 µM GST-KRAS- GMPPNP and -GDP flowed over NTA-immobilized BRAF-NT2 at a flow rate of 30 µl/min.

References

(1) Terrell, E. M.; Durrant, D. E.; Ritt, D. A.; Sealover, N. E.; Sheffels, E.; Spencer-Smith, R.; Esposito, D.; Zhou, Y.; Hancock, J. F.; Kortum, R. L.; Morrison, D. K. Distinct Binding Preferences between Ras and Raf Family Members and the Impact on Oncogenic Ras Signaling. Mol. Cell 2019, 76 (6), 872-884.e5. https://doi.org/10.1016/j.molcel.2019.09.004.

(2) Tran, T. H.; Chan, A. H.; Young, L. C.; Bindu, L.; Neale, C.; Messing, S.; Dharmaiah, S.; Taylor, T.; Denson, J. P.; Esposito, D.; Nissley, D. V.; Stephen, A. G.; McCormick, F.;Simanshu, D. K. KRAS Interaction with RAF1 RAS-Binding Domain and Cysteine-Rich Domain Provides Insights into RAS-Mediated RAF Activation. Nat. Commun. 2021, 12 (1176), 1–16. https://doi.org/10.1038/s41467-021-21422-x.

(3) Fischer, A.; Hekman, M.; Kuhlmann, J.; Rubio, I.; Wiese, S.; Rapp, U. R. B- and C-RAF Display Essential Differences in Their Binding to Ras: The Isotype-Specific N Terminus of B-RAF Facilitates Ras Binding. J. Biol. Chem. 2007, 282 (36), 26503–26516. https://doi.org/10.1074/jbc.M607458200.

(4) Park, E.; Rawson, S.; Li, K.; Kim, B. W.; Ficarro, S. B.; Pino, G. G. Del; Sharif, H.; Marto, J. A.; Jeon, H.; Eck, M. J. Architecture of Autoinhibited and Active BRAF–MEK1–14-3-3Complexes. Nature 2019, 575 (7783), 545–550. https://doi.org/10.1038/s41586-0191660-y.

(5) Tran, N. H.; Wu, X.; Frost, J. A. B-Raf and Raf-1 Are Regulated by Distinct Autoregulatory Mechanisms. J. Biol. Chem. 2005, 280 (16), 16244–16253. https://doi.org/10.1074/jbc.M501185200.

(6) Prior, I. A.; Hancock, J. F. Ras Trafficking, Localization and Compartmentalized Signalling. Semin. Cell Dev. Biol. 2012, 23 (2), 145–153.

(7) Herrmann, C.; Martin, G. A.; Wittinghofer, A. Quantitative Analysis of the Complex between P21 and the Ras-Binding Domain of the Human Raf-1 Protein Kinase. J. Biol. Chem. 1995, 270 (7), 2901–2905. https://doi.org/10.1074/jbc.270.7.2901.

(8) Vojtek, A. B.; Hollenberg, S. M.; Cooper, J. A. Mammalian Ras Interacts Directly with the Serine/Threonine Kinase Raf. Cell 1993, 74 (1), 205–214. https://doi.org/10.1016/00928674(93)90307-C.

(9) Parker, M. W.; Bello, M. Lo; Federici, G. Crystallization of Glutathione S-Transferase fromHuman Placenta. J. Mol. Biol. 1990, 213 (2), 221–222. https://doi.org/10.1016/S00222836(05)80183-4.

(10) Inouye, K.; Mizutani, S.; Koide, H.; Kaziro, Y. Formation of the Ras Dimer Is Essential for Raf-1 Activation. J. Biol. Chem. 2000, 275 (6), 3737–3740. https://doi.org/10.1074/JBC.275.6.3737.

(11) Z. Y. Kan, X. Ye, J. J. Skinner, L. Mayne, S. W. E. ExMS2: An Integrated Solution for Hydrogen-Deuterium Exchange Mass Spectrometry Data Analysis. Anal Chem 2019, 91 (11), 7474–7481.

(12) Mayne, L.; Kan, Z. Y.; Sevugan Chetty, P.; Ricciuti, A.; Walters, B. T.; Englander, S. W. Many Overlapping Peptides for Protein Hydrogen Exchange Experiments by the Fragment Separation-Mass Spectrometry Method. J. Am. Soc. Mass Spectrom. 2011, 22 (11), 1898–1905. https://doi.org/10.1007/S13361-011-0235-4.

(13) Ye, X.; Lin, J.; Mayne, L.; Shorter, J.; Englander, S. W. Hydrogen Exchange Reveals Hsp104 Architecture, Structural Dynamics, and Energetics in Physiological Solution.Proc. Natl. Acad. Sci. 2019, 116 (15), 7333–7342. https://doi.org/10.1073/pnas.1816184116.

(14) Ye, X.; Lin, J.; Mayne, L.; Shorter, J.; Englander, S. W. Structural and Kinetic Basis for the Regulation and Potentiation of Hsp104 Function. Proc. Natl. Acad. Sci. 2020, 117 (17), 9384–9392. https://doi.org/10.1073/pnas.1921968117.

(15) Hamuro, Y. Determination of Equine Cytochrome c Backbone Amide Hydrogen/Deuterium Exchange Rates by Mass Spectrometry Using a Wider Time Window and Isotope Envelope. J. Am. Soc. Mass Spectrom. 2017, 28 (3), 486–497. https://doi.org/10.1007/s13361-016-1571-1.

(16) Herrmann, C.; Horn, G.; Spaargaren, M.; Wittinghofer, A. Differential Interaction of theRas Family GTP-Binding Proteins H-Ras, Rap1A, and R-Ras with the Putative Effector Molecules Raf Kinase and Ral-Guanine Nucleotide Exchange Factor. J. Biol. Chem. 1996, 271 (12), 6794–6800. https://doi.org/10.1074/jbc.271.12.6794.

(17) Miller, A. F.; Halkides, C. J.; Redfield, A. G. An NMR Comparison of the Changes Produced by Different Guanosine 5’-Triphosphate Analogs in Wild-Type and Oncogenic Mutant P21ras. Biochemistry 1993, 32 (29), 7367–7376. https://doi.org/10.1021/bi00080a006.

(18) Amendola, C. R.; Mahaffey, J. P.; Parker, S. J.; Ahearn, I. M.; Chen, W. C.; Zhou, M.;Court, H.; Shi, J.; Mendoza, S. L.; Morten, M. J.; Rothenberg, E.; Gottlieb, E.; Wadghiri, Y. Z.; Possemato, R.; Hubbard, S. R.; Balmain, A.; Kimmelman, A. C.; Philips, M. R. KRAS4A Directly Regulates Hexokinase 1. Nature 2019. https://doi.org/10.1038/s41586019-1832-9.

(19) John, J.; Sohmen, R.; Feuerstein, J.; Linke, R.; Wittinghofer, A.; Goody, R. S. Kinetics of Interaction of Nucleotides with Nucleotide-Free H-Ras P21. Biochemistry 1990, 29 (25), 6058–6065. https://doi.org/10.1021/bi00477a025.

(20) Dharmaiah, S.; Tran, T. H.; Messing, S.; Agamasu, C.; Gillette, W. K.; Yan, W.;Waybright, T.; Alexander, P.; Esposito, D.; Nissley, D. V.; McCormick, F.; Stephen, A. G.; Simanshu, D. K. Structures of N-Terminally Processed KRAS Provide Insight into the Role of N-Acetylation. Sci. Reports 2019 91 2019, 9 (1), 1–15. https://doi.org/10.1038/s41598-019-46846-w.

(21) Rathinaswamy, M. K.; Gaieb, Z.; Fleming, K. D.; Borsari, C.; Harris, N. J.; Moeller, B. E.;Wymann, M. P.; Amaro, R. E.; Burke, J. E. Disease-Related Mutations in PI3Kγ Disrupt Regulatory C-Terminal Dynamics and Reveal a Path to Selective Inhibitors. Elife 2021, 10. https://doi.org/10.7554/eLife.64691.